# GSURE-Based Diffusion Model Training with Corrupted Data

**Bahjat Kawar**[*]                                                    *bahjat.kawar@cs.technion.ac.il*
*Department of Computer Science*

**Noam Elata**[*]                                                      *noamelata@campus.technion.ac.il*
*Department of Electrical and Computer Engineering*

**Tomer Michaeli**                                                     *tomer.m@ee.technion.ac.il*
*Department of Electrical and Computer Engineering*

**Michael Elad**                                                       *elad@cs.technion.ac.il*
*Department of Computer Science*

*Technion - Israel Institute of Technology, Haifa, Israel*

**Reviewed on OpenReview:** *https://openreview.net/forum?id=BRl7fqMwaJ*

## Abstract

Diffusion models have demonstrated impressive results in both data generation and downstream tasks such as inverse problems, text-based editing, classification, and more. However, training such models usually requires large amounts of clean signals which are often difficult or impossible to obtain. In this work, we propose a novel training technique for generative diffusion models based only on corrupted data. We introduce a loss function based on the Generalized Stein's Unbiased Risk Estimator (GSURE), and prove that under some conditions, it is equivalent to the training objective used in fully supervised diffusion models. We demonstrate our technique on face images as well as Magnetic Resonance Imaging (MRI), where the use of undersampled data significantly alleviates data collection costs. Our approach achieves generative performance comparable to its fully supervised counterpart without training on any clean signals. In addition, we deploy the resulting diffusion model in various downstream tasks beyond the degradation present in the training set, showcasing promising results[1].

## 1 Introduction

Denoising diffusion probabilistic models (DDPMs) (Sohl-Dickstein et al., 2015; Ho et al., 2020; Song & Ermon, 2019), or diffusion models for short, are a family of generative models that has recently risen to prominence. They have achieved state-of-the-art performance in image generation (Song et al., 2020b; Vahdat et al., 2021; Dhariwal & Nichol, 2021; Rombach et al., 2022; Kim et al., 2022), as well as impressive generative modeling capabilities in other modalities (Ho et al., 2022; Singer et al., 2023; Kong et al., 2021; Popov et al., 2021; Gong et al., 2023; Li et al., 2022; Tevet et al., 2022), including protein structures (Watson et al., 2022; Qiao et al., 2022; Schneuing et al., 2022; Yim et al., 2023) and medical data (Song et al., 2023; Chung & Ye, 2022; Jalal et al., 2021; Xie & Li, 2022; Chung et al., 2023; Li et al., 2023; Adib et al., 2023). The prowess and flexibility of DDPMs have enabled their profound impact on downstream applications (Kawar et al., 2022; Theis et al., 2022; Blau et al., 2022; Pinaya et al., 2022; Wyatt et al., 2022; Kawar et al., 2023; Zimmermann et al., 2021).

---

[*]Equal contribution.
[1]Our code is available at `https://github.com/bahjat-kawar/gsure-diffusion`.

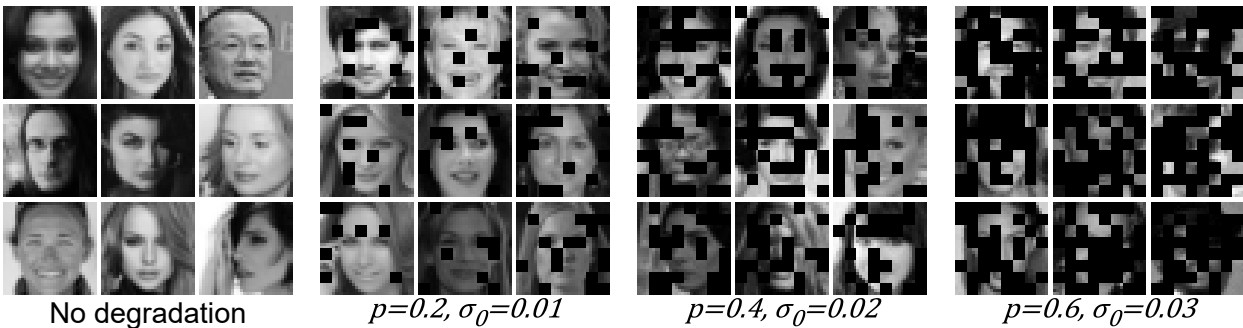

No degradation      $p=0.2, \sigma_0=0.01$      $p=0.4, \sigma_0=0.02$      $p=0.6, \sigma_0=0.03$

Figure 1: Training sets samples of the different degradation settings in CelebA (Liu et al., 2015) experiments.

Training a diffusion model to learn an unknown data distribution is a complex task. It usually requires training parameter-heavy neural networks on large amounts of pristine data. For instance, diffusion models' success in image generation was in part enabled by large curated datasets, containing millions or even billions of images (Deng et al., 2009; Schuhmann et al., 2022). However, such large-scale datasets of pristine samples may often be expensive, difficult, or even impossible to obtain, especially in the medical domain (Mullainathan & Obermeyer, 2022). Interest in the degraded data setting has risen in recent years Xiang et al. (2023); Aali et al. (2023); Daras et al. (2024), yet existing solutions address only specific cases. In this work, we present GSURE-Diffusion, a method for training generative diffusion models based on data corrupted by linear degradations and Gaussian noise. This setting can make data collection for deep learning significantly faster and less expensive.

GSURE-Diffusion operates on a datasets of noisy linear measurements of signals, and assumes the signal acquisition process is randomized within a fixed general structure, which is the case in many real-world applications. In this setting, we present a novel loss function to learn the underlying data distribution. First, we use the Singular Value Decomposition (SVD) of the degradation operators to decouple the measurement equation, following DDRM (Kawar et al., 2022). This transformation simplifies the degradation into a masking operation. Then, we add synthetic noise to the SVD-transformed measurements, likening them to the noisy samples used in the DDPM (Ho et al., 2020) framework. Finally, we use the ensemble version of the Generalized Stein's Unbiased Risk Estimator (GSURE) (Aggarwal et al., 2022; Eldar, 2008) to learn to denoise samples without access to ground-truth clean signals. Simply put, GSURE's mathematical formulation allows us to makes use of the undamaged data within the corrupted image. Our proposed GSURE-based loss function is general to all randomized linear measurement settings, and we prove its equivalence to the fully supervised denoising diffusion loss under some conditions.

To empirically evaluate our technique, we apply it on a downsized grayscale version of CelebA (Liu et al., 2015), a dataset of celebrity face images. We train a GSURE-Diffusion model on noisy images with randomly missing patches, and compare its generative output with an oracle model that trained on the full clean images. We observe that GSURE-Diffusion results in comparable generative performance to the oracle, despite having trained solely on corrupted data.

Furthermore, we use GSURE-Diffusion for Magnetic Resonance Imaging (MRI), a ubiquitous medical imaging modality providing vital diagnostic information. Generative models for MRI are usually trained on datasets of fully sampled MR images, which can be expensive to obtain. In contrast, we train a generative model on noisy undersampled data obtained from accelerated MRI scans. By utilizing GSURE-Diffusion, we train a model with performance comparable to an oracle version while significantly reducing the time and resources required to collect the training dataset. Then, we showcase the flexibility of the generative diffusion model we obtain. We use this model for accelerated MRI reconstruction, extending its applicability to acceleration factors beyond those encountered during training. Moreover, this model may serve as a foundational framework for addressing various tasks. We demonstrate its capabilities in MRI reconstruction for different subsampling strategies and uncertainty quantification, and compare its results with an oracle version.

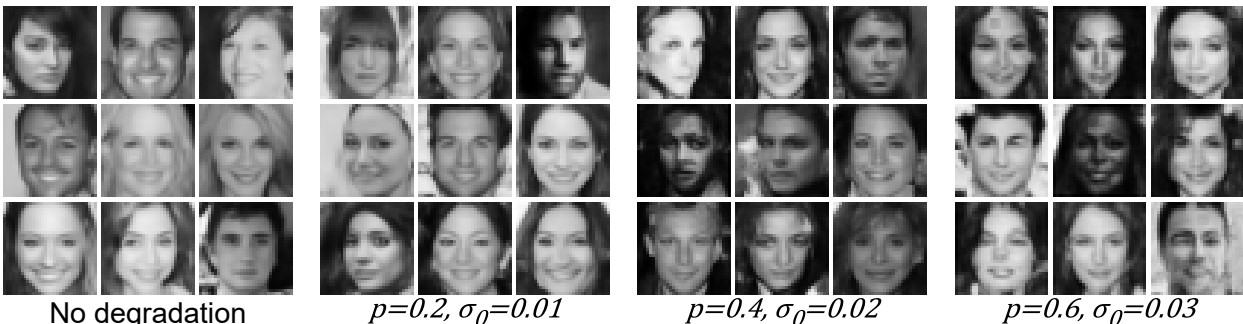

| No degradation | $p=0.2, \sigma_0=0.01$ | $p=0.4, \sigma_0=0.02$ | $p=0.6, \sigma_0=0.03$ |

Figure 2: Generated samples (with 50 DDIM (Song et al., 2020a) steps) from models trained on different degradation settings in CelebA (Liu et al., 2015) experiments.

In conclusion, we present GSURE-Diffusion, a novel method for training generative diffusion models based on corrupted measurements of the underlying data. We demonstrate the capabilities of the method through several experiments, and show its applicability to real-world problems. We hope that GSURE-Diffusion will facilitate future work on generative modelling for challenging settings, generalizing for more complex scenarios and various modalities.

## 2 Background

### 2.1 Denoising Diffusion Probabilistic Models

Denoising Diffusion Probabilistic Models (DDPMs) (Ho et al., 2020) are a family of generative models that learn a distribution $p_\theta(\mathbf{x})$, approximating a data distribution $q(\mathbf{x})$ from a dataset $\mathcal{D}$ of samples. DDPMs follow a Markov chain structure $\mathbf{x}_T \to \mathbf{x}_{T-1} \to \cdots \to \mathbf{x}_1 \to \mathbf{x}_0$ that reverses a forward noising process from $\mathbf{x}_0$ to $\mathbf{x}_T$. In the forward process, $\mathbf{x}_0$ is set to be $\mathbf{x}$, and the intermediate variables $\mathbf{x}_t$ are defined by $q^{(t)}(\mathbf{x}_t|\mathbf{x}_{t-1})$, usually chosen to be a simple Gaussian $\mathcal{N}\left(\sqrt{1-\beta_t}\mathbf{x}_{t-1}, \beta_t \boldsymbol{I}\right)$. This leads to a useful property, $q^*(\mathbf{x}_t|\mathbf{x}_0) = \mathcal{N}\left(\sqrt{\bar{\alpha}_t}\mathbf{x}_0, (1-\bar{\alpha}_t)\boldsymbol{I}\right)$ with $\bar{\alpha}_t = \prod_{s=1}^{t}(1-\beta_t)$, which facilitates model training. In the reverse and more challenging process, the learned distribution $p_\theta^{(t)}(\mathbf{x}_{t-1}|\mathbf{x}_t)$ is also modeled as Gaussian, with a learned mean dependent on a neural network $f_\theta^{(t)}(\mathbf{x}_t)$ and a fixed (Ho et al., 2020) or learned (Nichol & Dhariwal, 2021) covariance.

The diffusion model $f_\theta^{(t)}(\mathbf{x}_t)$ is trained to optimize an evidence lower bound (ELBO) on the log-likelihood objective (Sohl-Dickstein et al., 2015). The ELBO can be simplified into the following denoising objective:

$$\sum_{t=1}^{T} \gamma_t \mathbb{E}\left[\left\|f_\theta^{(t)}(\mathbf{x}_t) - \mathbf{x}_0\right\|_2^2\right], \tag{1}$$

where the $\gamma_t$ assign weights to different $t$ (different noise levels), and the expectation is over some $\mathbf{x}_t \sim q(\mathbf{x}_t|\mathbf{x}_0)$, $\mathbf{x}_0 \sim q(\mathbf{x})$. Please refer to (Ho et al., 2020; Song et al., 2020a) for derivations. After training, diffusion models synthesize data by starting with a sample $\mathbf{x}_T \sim \mathcal{N}(0, \boldsymbol{I})$, following the learned distributions $p_\theta^{(t)}$ along the Markov chain, sampling from each, and outputting $\mathbf{x}_0$ as the final sample. Diffusion models have had incredible success in image generation (Song et al., 2020b; Vahdat et al., 2021; Dhariwal & Nichol, 2021; Rombach et al., 2022; Kim et al., 2022), as well as generation of other modalities (Ho et al., 2022; Singer et al., 2023; Kong et al., 2021; Popov et al., 2021; Gong et al., 2023; Li et al., 2022; Tevet et al., 2022). They have also been deployed in a myriad of related tasks (Kawar et al., 2022; Theis et al., 2022; Blau et al., 2022; Pinaya et al., 2022; Wyatt et al., 2022; Kawar et al., 2023; Zimmermann et al., 2021). In this work, we revisit DDPMs and seek a way to train them using only corrupted data.

## 2.2 Generalized Stein's Unbiased Risk Estimator (GSURE)

Given noisy measurements $\mathbf{y} = \mathbf{x} + \mathbf{z}$ (where $\mathbf{x}, \mathbf{y}, \mathbf{z} \in \mathbb{R}^n$) with noise $\mathbf{z} \sim \mathcal{N}(0, \sigma^2 \boldsymbol{I})$, and a function $f(\mathbf{y})$ aiming to estimate $\mathbf{x}$ from $\mathbf{y}$, Stein's unbiased risk estimator (SURE) (Stein, 1981) is an unbiased estimator for the mean squared error (MSE) of $f(\mathbf{y})$, formulated as

$$\mathbb{E}\left[\|f(\mathbf{y}) - \mathbf{x}\|_2^2\right] = \mathbb{E}\left[\|f(\mathbf{y}) - \mathbf{y}\|_2^2\right] + 2\sigma^2 \mathbb{E}\left[\nabla_{\mathbf{y}} \cdot f(\mathbf{y})\right] - n\sigma^2. \tag{2}$$

Crucially, SURE provides the ability to estimate the MSE of a denoiser $f(\mathbf{y})$ without access to clean signals $\mathbf{x}$. As a result, many researchers have used SURE for unsupervised learning of denoisers (Zhang & Desai, 1998; Blu & Luisier, 2007; Metzler et al., 2018; Soltanayev & Chun, 2018; Nguyen et al., 2020; Jo et al., 2021), even extending to training diffusion models based on (fully sampled) noisy data (Xiang et al., 2023; Aali et al., 2023).

In the context of inverse problems, SURE has been generalized for corrupted measurements beyond additive white Gaussian noise (Eldar, 2008). The Generalized SURE (GSURE) considers $\mathbf{y} = \boldsymbol{H}\mathbf{x} + \mathbf{z}$ (where $\mathbf{x} \in \mathbb{R}^n$, $\boldsymbol{H} \in \mathbb{R}^{m \times n}$, $\mathbf{y}, \mathbf{z} \in \mathbb{R}^m$, and $\mathbf{z} \sim \mathcal{N}(0, \boldsymbol{C})$), and a function $f(\mathbf{y})$ estimating $\mathbf{x}$. In this case, GSURE provides an unbiased estimate for the projected MSE:

$$\mathbb{E}\left[\|\boldsymbol{P}\left(f(\mathbf{y}) - \mathbf{x}\right)\|_2^2\right] = \mathbb{E}\left[\|\boldsymbol{P}\left(f(\mathbf{y}) - \mathbf{x}_{\mathrm{ML}}\right)\|_2^2\right] + 2\mathbb{E}\left[\nabla_{\boldsymbol{H}^\top \boldsymbol{C}^{-1} \mathbf{y}} \cdot \boldsymbol{P} f(\mathbf{y})\right] + c, \tag{3}$$

where $\boldsymbol{H}^\dagger$ is the Moore-Penrose pseudo-inverse of $\boldsymbol{H}$, $\boldsymbol{P} = \boldsymbol{H}^\dagger \boldsymbol{H}$ is a projection matrix onto the range-space of $\boldsymbol{H}$, $\mathbf{x}_{\mathrm{ML}} = \left(\boldsymbol{H}^\top \boldsymbol{C}^{-1} \boldsymbol{H}\right)^\dagger \boldsymbol{H}^\top \boldsymbol{C}^{-1} \mathbf{y}$, and $c$ is a constant that does not depend on $f(\mathbf{y})$. Several works have utilized GSURE for solving inverse problems by training only on corrupted measurements (Metzler et al., 2018; Zhussip et al., 2019; Liu et al., 2020; Abu-Hussein et al., 2022). However, when $\boldsymbol{H}$ causes significant information loss, the projected MSE stops being a good proxy for the full MSE. Ensemble SURE (ENSURE) (Aggarwal et al., 2022) learns from a dataset of measurements, each corrupted by a different operator $\boldsymbol{H}$. Therefore, the expectation over the projected MSE is taken over $\boldsymbol{H}$ as well as the data and noise. This constitutes a more accurate proxy for the full MSE without relying on clean signals. In this work, we extend the ENSURE framework for training a diffusion model using corrupted data.

## 3 GSURE-Diffusion: Mathematical Formulation

### 3.1 Problem Formulation

We are interested in training a generative diffusion model that can sample from an unknown data distribution $q(\mathbf{x})$. However, we only have access to a dataset $\mathcal{D}$ of corrupted measurements

$$\mathbf{y} = \boldsymbol{H}\mathbf{x} + \mathbf{z}, \tag{4}$$

where $\mathbf{y} \in \mathbb{R}^m$, $\mathbf{x} \in \mathbb{R}^n$, $\boldsymbol{H} \in \mathbb{R}^{m \times n}$, and $\mathbf{z} \sim \mathcal{N}(0, \sigma_0^2 \boldsymbol{I})$ is additive white Gaussian noise (AWGN).[2] Equation 4 refers to a single instance of an ideal image and its corresponding measurement, and more generally, different measurements $\mathbf{y}$ in the dataset may relate to different signals $\mathbf{x}$, different degradation procedures $\boldsymbol{H}$, and different noise realizations $\mathbf{z}$. We assume $\mathbf{x}$, $\mathbf{z}$, and $\boldsymbol{H}$ are randomly and independently sampled from their respective distributions.

In order to decouple the mathematical relationship between the observed measurements and the underlying data, we follow (Kawar et al., 2022) and utilize the singular value decomposition (SVD) of $\boldsymbol{H}$,

$$\boldsymbol{H} = \boldsymbol{U}\boldsymbol{\Sigma}\boldsymbol{V}^\top, \tag{5}$$

where $\boldsymbol{U} \in \mathbb{R}^{m \times m}$ and $\boldsymbol{V} \in \mathbb{R}^{n \times n}$ are orthogonal matrices, and $\boldsymbol{\Sigma} \in \mathbb{R}^{m \times n}$ is a rectangular diagonal matrix containing the singular values of $\boldsymbol{H}$. We define $\bar{\mathbf{x}} = \boldsymbol{V}^\top \mathbf{x}$, $\bar{\mathbf{y}} = \boldsymbol{\Sigma}^\dagger \boldsymbol{U}^\top \mathbf{y}$, and $\bar{\mathbf{z}} = \boldsymbol{\Sigma}^\dagger \boldsymbol{U}^\top \mathbf{z}$. Using these definitions and the SVD, Equation 4 becomes

$$\bar{\mathbf{y}} = \boldsymbol{P}\bar{\mathbf{x}} + \bar{\mathbf{z}}, \tag{6}$$

---

[2]Our method can also handle anisotropic uncorrelated noise. We only consider AWGN to simplify notations.

where $\boldsymbol{P} = \boldsymbol{\Sigma}^\dagger \boldsymbol{\Sigma}$ is a diagonal subsampling matrix with zeroes and ones, and $\bar{\mathbf{z}} \sim \mathcal{N}(0, \sigma_0^2 \boldsymbol{\Sigma}^\dagger \boldsymbol{\Sigma}^{\dagger\top})$ constitutes anisotropic uncorrelated Gaussian noise.

We make the following assumptions on the training dataset $\mathcal{D}$: (i) The sampling matrices $\boldsymbol{H}$ and noise levels $\sigma_0$ are known; (ii) All matrices $\boldsymbol{H}$ share the same right-singular vectors $\boldsymbol{V}^\top$; and (iii) The different $\boldsymbol{H}$ across the dataset jointly cover the signal space $\mathbb{R}^n$, *i.e.*, $\mathbb{E}[\boldsymbol{P}]$ is a positive definite matrix.[3] These assumptions are satisfied in many real-world applications such as Magnetic Resonance Imaging (MRI) – measurements are acquired in a similar fashion for all data points (upholding (ii)), and the subsampling pattern can be randomly chosen for each point (upholding (iii)). $\boldsymbol{H}$ and $\sigma_0$ are derived from the physics of the signal acquisition procedure, thus upholding (i).

Under the transformed measurement equation presented in Equation 6, we aim to train a generative model for $\bar{\mathbf{x}}$, which can easily translate to $\mathbf{x}$ through the orthogonal transformation $\mathbf{x} = \boldsymbol{V}\bar{\mathbf{x}}$.

### 3.2 GSURE-Based Denoising Diffusion Loss Function

In order to train a diffusion model for $\bar{\mathbf{x}}$, we aim to obtain noisy training samples $\bar{\mathbf{x}}_t$ that satisfy the marginal distribution $q^*(\bar{\mathbf{x}}_t|\bar{\mathbf{x}}) = \mathcal{N}\left(\sqrt{\bar{\alpha}_t}\bar{\mathbf{x}}, (1-\bar{\alpha}_t)\boldsymbol{I}\right)$, as in traditional diffusion models. However, we only have access to corrupted measurements $\bar{\mathbf{y}}$ as in Equation 6. For a given $t$, we perturb these measurements with additional noise according to

$$\bar{\mathbf{x}}_t = \sqrt{\bar{\alpha}_t}\bar{\mathbf{y}} + \left((1-\bar{\alpha}_t)\boldsymbol{I} - \bar{\alpha}_t\sigma_0^2\boldsymbol{\Sigma}^\dagger\boldsymbol{\Sigma}^{\dagger\top}\right)^{\frac{1}{2}}\boldsymbol{\epsilon}_t, \tag{7}$$

where $\boldsymbol{\epsilon}_t \sim \mathcal{N}(0, \boldsymbol{I})$ is independently sampled. Intuitively, $\sqrt{\bar{\alpha}_t}\bar{\mathbf{y}}$ includes noise with a diagonal covariance $\bar{\alpha}_t\sigma_0^2\boldsymbol{\Sigma}^\dagger\boldsymbol{\Sigma}^{\dagger\top}$. We increase the noise level in each entry by an appropriate amount to reach a variance of $1-\bar{\alpha}_t$ in all entries. Increasing the noise to the desired level is possible as long as $\left((1-\bar{\alpha}_t)\boldsymbol{I} - \bar{\alpha}_t\sigma_0^2\boldsymbol{\Sigma}^\dagger\boldsymbol{\Sigma}^{\dagger\top}\right)$ is a positive-semi-definite (PSD) matrix. Because $\bar{\alpha}_t$ is monotonically decreasing w.r.t $t$, by setting the beginning of the noise schedule such that $\left((1-\bar{\alpha}_t)\boldsymbol{I} - \bar{\alpha}_t\sigma_0^2\boldsymbol{\Sigma}^\dagger\boldsymbol{\Sigma}^{\dagger\top}\right)$ is PSD, we ensure that this covariance matrix is PSD for all timesteps. This way, we obtain samples $\bar{\mathbf{x}}_t$ suitable for training a diffusion model, as they follow the marginal distribution

$$q(\bar{\mathbf{x}}_t|\bar{\mathbf{x}}, \boldsymbol{P}) = \mathcal{N}\left(\sqrt{\bar{\alpha}_t}\boldsymbol{P}\bar{\mathbf{x}}, (1-\bar{\alpha}_t)\boldsymbol{I}\right). \tag{8}$$

This resembles the ideal distribution of training samples $q^*(\bar{\mathbf{x}}_t|\bar{\mathbf{x}})$, differing only in the mean value for entries dropped by $\boldsymbol{P}$. In the following, we derive a loss function that uses $\bar{\mathbf{x}}_t$ satisfying Equation 8, and utilizes an expectation over $\bar{\mathbf{x}}$, $\bar{\mathbf{z}}$, and $\boldsymbol{P}$ yielding samples $\bar{\mathbf{x}}_t \sim \mathbb{E}_{\boldsymbol{P}}[q(\bar{\mathbf{x}}_t|\bar{\mathbf{x}}, \boldsymbol{P})]$. This results in an estimate for denoising ideal samples from $q^*(\bar{\mathbf{x}}_t|\bar{\mathbf{x}})$. The estimate assumes that the model has the ability to generalize for samples from $q^*(\bar{\mathbf{x}}_t|\bar{\mathbf{x}}) = q(\bar{\mathbf{x}}_t|\bar{\mathbf{x}}, \boldsymbol{P} = \boldsymbol{I})$, despite having trained only on signals with an undersampled $\boldsymbol{P}$. We validate our model's ability to denoise samples from $q^*(\bar{\mathbf{x}}_t|\bar{\mathbf{x}})$ in subsection D.3.

Ideally, we would like to train a diffusion model $f_\theta^{(t)}(\bar{\mathbf{x}}_t)$ using the traditional denoising diffusion loss function in Equation 1. However, as we only have access to undersampled measurements, we consider a weighted expected projected MSE objective (similar to ENSURE (Aggarwal et al., 2022)):

$$\sum_{t=1}^{T} \gamma_t \mathbb{E}\left[\left\|\boldsymbol{W}\boldsymbol{P}\left(f_\theta^{(t)}(\bar{\mathbf{x}}_t) - \bar{\mathbf{x}}\right)\right\|_2^2\right], \tag{9}$$

where the expectation is taken over $\bar{\mathbf{x}}_t \sim q(\bar{\mathbf{x}}_t|\bar{\mathbf{x}}, \boldsymbol{P})$, $\bar{\mathbf{x}} \sim q(\bar{\mathbf{x}})$, $\boldsymbol{P}$ is independently sampled and $\boldsymbol{W} = \mathbb{E}[\boldsymbol{P}]^{-\frac{1}{2}} \succ 0$ (positive definite). The weight matrix $\mathbf{W}$ is placed in Equation 9 for balancing the effect of the projections $\mathbf{P}$, in the case where elements of $\mathbf{P}$ do not have equal probability. In practice, this expectation is realized through $\bar{\mathbf{y}}$ sampled from the dataset $\mathcal{D}$ and Equation 7.

**Proposition 3.1.** *For $\mathbf{x} \sim q(\mathbf{x})$, $\bar{\mathbf{x}} = \boldsymbol{V}^\top\mathbf{x}$, $\bar{\mathbf{x}}_t$ sampled from Equation 8, and the diagonal weight matrix $\boldsymbol{W} = \mathbb{E}[\boldsymbol{P}]^{-\frac{1}{2}} \succ 0$ (positive definite), if $\boldsymbol{P}$ and $\left(f_\theta^{(t)}(\bar{\mathbf{x}}_t) - \bar{\mathbf{x}}\right)$ are statistically independent, then Equation 9 equals Equation 1.*

---

[3]Under these notations, $\mathbb{E}[\boldsymbol{P}]$ is measured for a fixed $\boldsymbol{V}^\top$, and the values in $\boldsymbol{\Sigma}$ are not necessarily ordered.

We place the proof in Appendix A. The proof relies on the aforementioned assumptions made in ENSURE (Aggarwal et al., 2022). We assess the validity of this assumption for our trained networks in subsection D.4, and find that the assumption approximately holds for half the range of timesteps used in the denoiser. Nevertheless, even though the assumption is violated for some timesteps, the algorithm's performance remains similar to the oracle model, as shown in Figure 4. The expected projected MSE term in Equation 9 measures the squared error of the denoiser $f_\theta^{(t)}(\bar{\mathbf{x}}_t)$ only in entries kept by $\boldsymbol{P}$. This fact makes the loss easier to measure, as we do not have access to the entries dropped by $\boldsymbol{P}$. However, we still cannot accurately measure this loss, because we lack access to noiseless signals $\boldsymbol{P}\bar{\mathbf{x}}$. To mitigate this, we utilize GSURE to estimate Equation 9 using only $\bar{\mathbf{x}}_t$ with the loss

$$\sum_{t=1}^{T} \gamma_t \mathbb{E}\left[\left\|\boldsymbol{W}\boldsymbol{P}\left(f_\theta^{(t)}(\bar{\mathbf{x}}_t) - \frac{1}{\sqrt{\bar{\alpha}_t}}\bar{\mathbf{x}}_t\right)\right\|_2^2 + 2\lambda_t\left(\nabla_{\bar{\mathbf{x}}_t} \cdot \boldsymbol{P}\boldsymbol{W}^2 f_\theta^{(t)}(\bar{\mathbf{x}}_t)\right) + c\right], \tag{10}$$

where $c$ is a constant that does not depend on $\theta$, and $\lambda_t$ is a scalar hyperparameter. The expectation is over the same random variables from Equation 9.

**Proposition 3.2.** *For $\mathbf{x} \sim q(\mathbf{x})$, $\bar{\mathbf{x}} = \boldsymbol{V}^\top \mathbf{x}$, $\bar{\mathbf{x}}_t$ sampled from Equation 8, and $\lambda_t = 1 - \bar{\alpha}_t$, it holds that Equation 10 equals Equation 9.*

We place the proof in the Appendix A. Proposition 3.1 and Proposition 3.2 present a principled method to train a denoising diffusion model based only on corrupted data $\bar{\mathbf{y}}$. By minimizing the loss function in Equation 10, we obtain a trained diffusion model that can be utilized in the same fashion as a fully supervised one, for both generation and downstream applications.

When our proposed training scheme is applied in practice, we apply the following modifications for practical considerations: First, the expectation term in Equation 10 is replaced by an average over training batches. Second, $\frac{1}{\sqrt{\bar{\alpha}_t}}\bar{\mathbf{x}}_t$ is replaced by $\bar{\mathbf{y}}$ to alleviate the high variance of the loss function at little to no cost in terms of bias. Lastly, the divergence term is calculated using the Hutchinson's trace estimator, which is an unbiased Monte Carlo estimator (Ramani et al., 2008; Hutchinson, 1989). While this estimator introduces some noise into our equation, we find that the estimation error is negligible when using a numerically stable derivative calculation. We expand upon these pragmatic implementation details in Appendix D.

## 4    Experiments

In the following, we demonstrate the capabilities of our method for training a diffusion model using corrupted data. To obtain corrupted data, we simulate several corruptions on datasets containing clean images. Then, we train a diffusion model based only on the corrupted data, and compare its results against an *oracle* model (with identical training hyperparameters) which is trained on the pristine data with the traditional diffusion loss function from Equation 1.

Table 1: FID (Heusel et al., 2017) results for diffusion models trained on increasing levels of degradation for $32 \times 32$-pixel CelebA (Liu et al., 2015) images, with different DDIM (Song et al., 2020a) steps at generation time. Models were trained with (top) or without (bottom) GSURE-Diffusion, on degraded data.

| Training Scheme | Data Degradation | 10 Steps | 20 Steps | 50 Steps | 100 Steps |
|---|---|---|---|---|---|
| Regular | No degradation (oracle) | 21.99 | 13.09 | 08.15 | 06.84 |
| | $p = 0.2$, $\sigma_0 = 0.01$ | 86.25 | 108.56 | 119.30 | 123.08 |
| | $p = 0.4$, $\sigma_0 = 0.02$ | 216.74 | 229.28 | 236.61 | 240.03 |
| | $p = 0.6$, $\sigma_0 = 0.03$ | 273.62 | 279.17 | 280.89 | 281.60 |
| GSURE-Diffusion | $p = 0.2$, $\sigma_0 = 0.01$ | 18.77 | 12.25 | 08.84 | 08.82 |
| | $p = 0.4$, $\sigma_0 = 0.02$ | 19.26 | 14.98 | 14.03 | 15.14 |
| | $p = 0.6$, $\sigma_0 = 0.03$ | 34.51 | 27.74 | 26.42 | 28.31 |

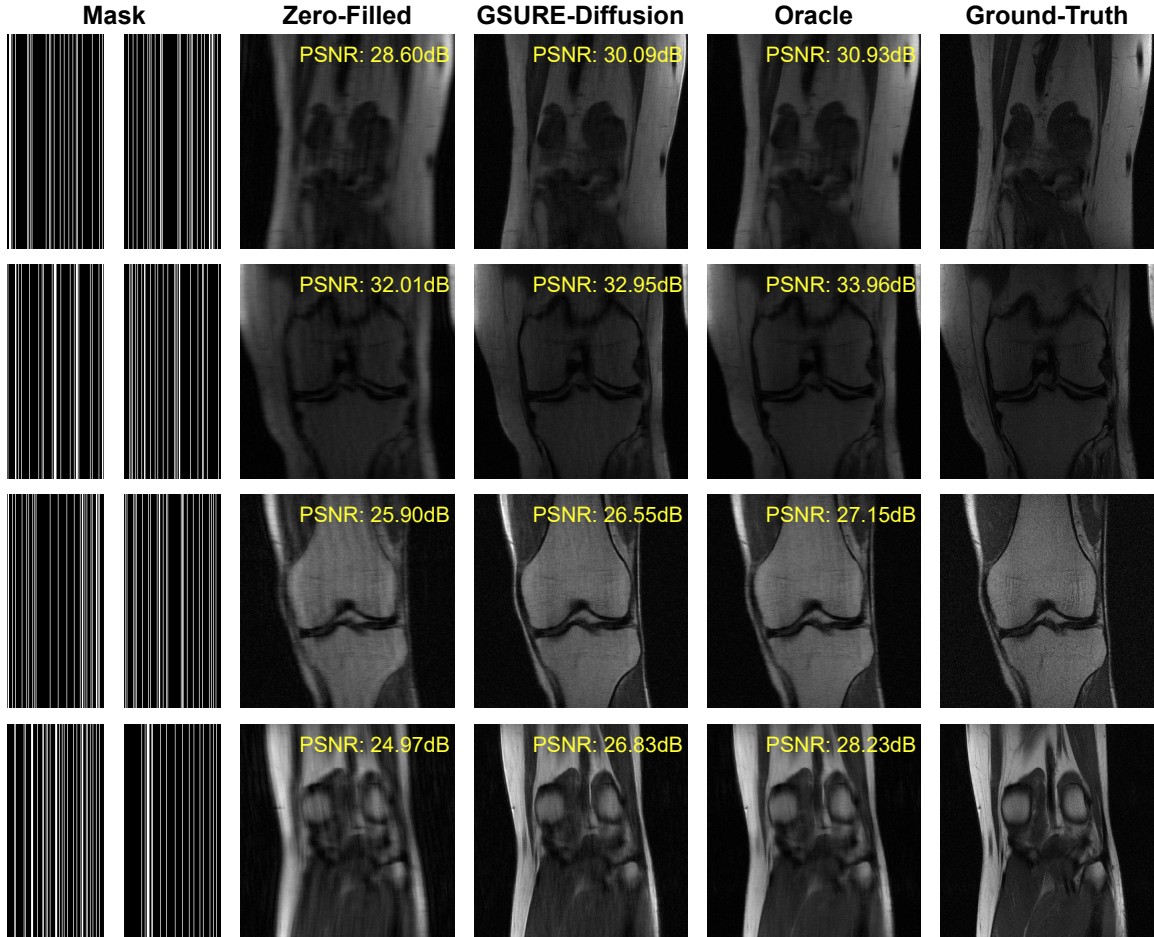

Figure 3: Accelerated MRI reconstruction results for $R = 4$ and $\sigma_0 = 0.01$.

## 4.1 Human Face Images

To empirically evaluate GSURE-Diffusion, we apply it on $32 \times 32$-pixel grayscale face images from CelebA (Liu et al., 2015). We simulate a corrupted measurement process by splitting the images into $4 \times 4$-pixel non-overlapping patches, and randomly erasing each patch with probability $p$. We also perturb the data with AWGN with standard deviation $\sigma_0$. This degradation matches our assumptions in subsection 3.1 (see Appendix B), and we provide samples of it in Figure 1. We adapt the U-Net (Ronneberger et al., 2015) architecture from DDPM (Ho et al., 2020) to match the image dimensions, and train diffusion models for increasing levels of degradation on the CelebA training set. Training hyperparameters and more details are provided in Appendix C.

After training, we generate images from the models using the deterministic DDIM (Song et al., 2020a) sampling schedule. We measure the generative performance using the Fréchet Inception Distance (FID) (Heusel et al., 2017) between 10000 generated images and the CelebA validation set. As can be seen in Table 1 and Figure 2, our GSURE-Diffusion models achieve generative performance comparable to the oracle model, despite having trained only on corrupted data. As expected, the performance deteriorates when more severe degradations are applied to the training data. We note that in the few timesteps regime, GSURE-Diffusion slightly outperforms the oracle model, possibly due to the regularization effect of training on corrupted measurements. However, once we increase the number of timesteps used for generation, the oracle model benefits from the more fine-grained process, whereas GSURE-Diffusion struggles to do so.

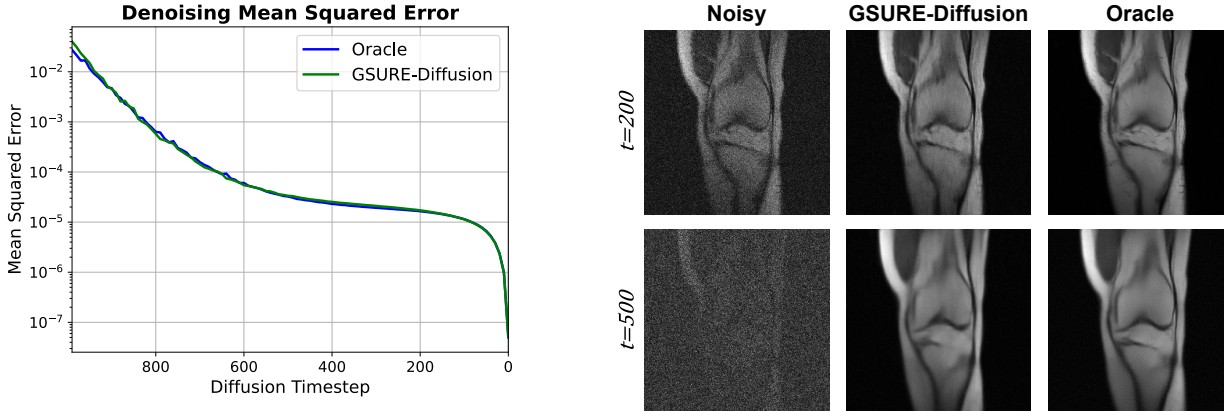

Figure 4: Left: Denoising MSE (on fully sampled noisy images) for the GSURE-Diffusion and oracle models across diffusion timesteps. Right: Qualitative denoising examples for both models.

## 4.2 Magnetic Resonance Imaging (MRI)

Magnetic Resonance Imaging (MRI) is a ubiquitous non-invasive medical imaging modality that can provide life-saving diagnostic information. MRI measurements are obtained in the Fourier spectrum (also called k-space) of an object with magnetic fields. However, measuring the entire k-space can be time-consuming and expensive. Therefore, inferring the scanned object image based on partial, possibly randomized and noisy, k-space measurements from accelerated MRI scans is a highly relevant and challenging inverse problem, drawing considerable research attention (Wang et al., 2016; Hammernik et al., 2018; Lee et al., 2018; Han et al., 2019; Weiss et al., 2021; Wang et al., 2022).

The accelerated MRI procedure satisfies the assumptions we introduced in subsection 3.1 (see Appendix B), with the discrete Fourier transform as $\boldsymbol{V}^{\top}$, and the random subsampling mask as $\boldsymbol{\Sigma}$. We use this fact and train a generative diffusion model for MRI based solely on accelerated scans. We train on $24,853$ scanned slices from the fastMRI (Knoll et al., 2020; Zbontar et al., 2019) single-coil knee MRI dataset, center-cropped to a spatial size of $320 \times 320$. The accelerated MRI subsampling process is simulated following (Jalal et al., 2021) with an acceleration factor $R = 4$, randomized subsampling of high frequencies, and AWGN with $\sigma_0 = 0.01$. To facilitate training, real and imaginary elements of complex numbers are treated as separate input/output channels. We use the same U-Net (Ronneberger et al., 2015; Ho et al., 2020) architecture from our CelebA (Liu et al., 2015) experiments, slightly modified to match the data dimensions, and train a diffusion model on the corrupted measurements. A separate oracle model is trained with the same hyperparameters (detailed in Appendix C) on the fully sampled data.

To evaluate the validity of our approach, we measure the mean squared error (MSE) of both models in denoising 1024 fully sampled MR images from the fastMRI (Knoll et al., 2020; Zbontar et al., 2019) validation set for different diffusion timesteps. In Figure 4, we observe that the denoising ability of GSURE-Diffusion resembles that of the oracle model. This supports our claim that GSURE-Diffusion can be a suitable replacement for the oracle model despite having trained exclusively on corrupted data. To test this claim in real-world applications, we substitute the oracle model for its GSURE-Diffusion counterpart in accelerated MRI reconstruction – restoring MR images from noisy subsampled versions. We use the general-purpose diffusion-based inverse problem solver DDRM (Kawar et al., 2022) for this task with $\eta = 0$ and 100 steps. In Figure 3, we observe that GSURE-Diffusion achieves similar performance to the oracle for $R = 4$. In all MRI experiments, we also present the "zero-filled" result (showing the MR image with zeroes in the missing frequencies) as a baseline.

Moreover, since we train a foundational MRI generative model, it is not restricted to the degradation setting present in its training data. For instance, the model can generalize well for higher acceleration factors, as we show in Figure 5. GSURE-Diffusion can also be utilized to reconstruct MR images corrupted by subsampling masks of different characteristics, such as 1-dimensional Gaussian random sampling and variable density

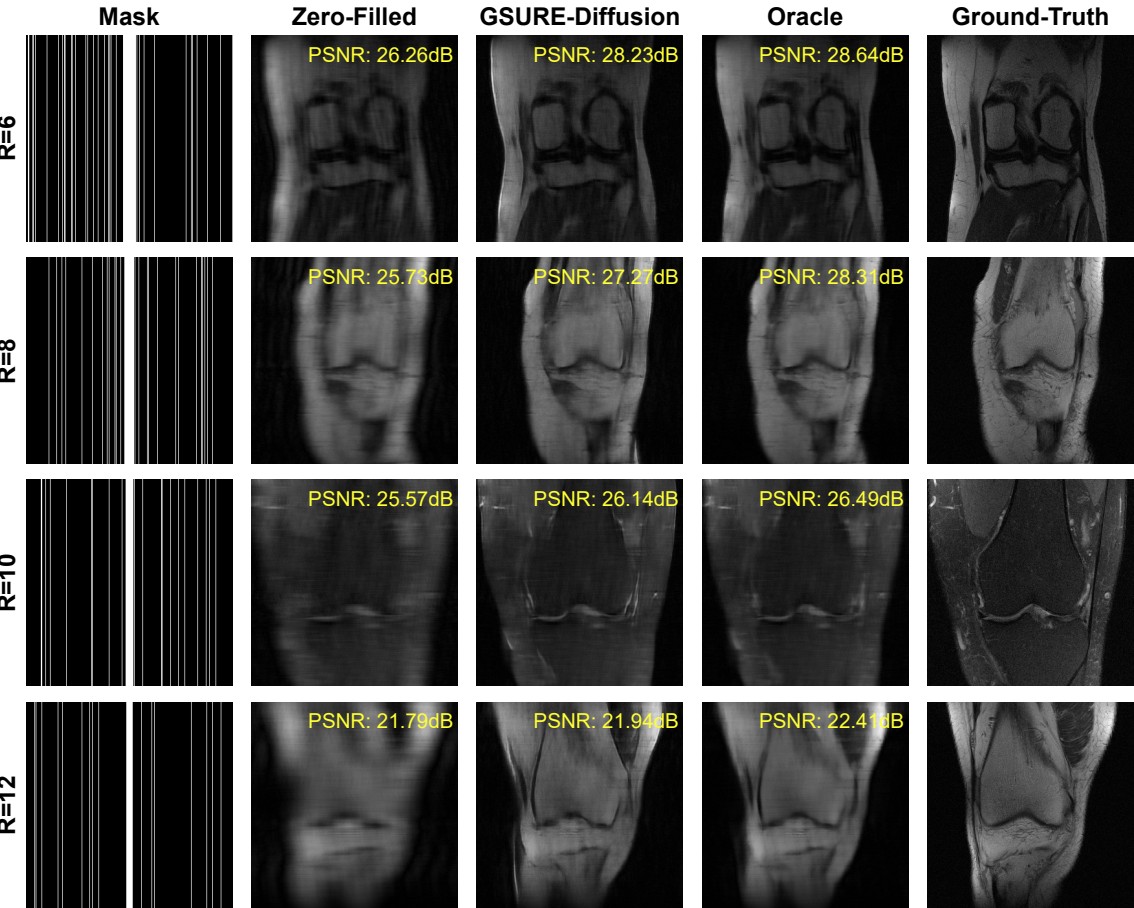

Figure 5: Accelerated MRI reconstruction results for $R \in \{6, 8, 10, 12\}$ and $\sigma_0 = 0.01$. GSURE-Diffusion can generalize well across different acceleration factors.

Poisson disc sampling, as we show in Figure 6. Furthermore, as a generative model, GSURE-Diffusion can provide uncertainty estimates for its outputs. We follow (Chung & Ye, 2022) and quantify the uncertainty using the standard deviation of 8 stochastic outputs made by the model. We add synthetic Gaussian noise to MR images with $\sigma_0 = 0.4$, and show uncertainty quantification results using GSURE-Diffusion and the oracle model in Figure 7. This uncertainty quantification technique can potentially aid medical practitioners, providing clues towards anomalous regions in MRI scans. These results present evidence that a generative model trained on corrupted data can be deployed in various applications. By loosening the requirements on the quality of the training data, we significantly reduce the cost of data acquisition for model training.

## 5    Limitations

While GSURE-Diffusion achieves impressive results, it suffers from a few limitations. First, the assumptions over the available dataset and its acquisition procedure listed in subsection 3.1 and subsection 3.2, which are satisfied in our example for accelerated MRI, may not always hold in other real-world scenarios. Second, if the measurement noise $\bar{\mathbf{z}}$ has high variance in at least one entry, the minimum noise level in the diffusion process, $\bar{\alpha}_1$, would also need to be high. This may lead to poor performance, as the final step of the generative diffusion process will need to clean significant noise. Third, if the distribution of $\boldsymbol{P}$ in the dataset is heavily biased towards certain regions, or has high variance in terms of $\boldsymbol{P}$'s rank, this can break the diffusion model's ability to infer $\boldsymbol{P}$ from $\bar{\mathbf{x}}_t$, and thus degrade the generative performance.

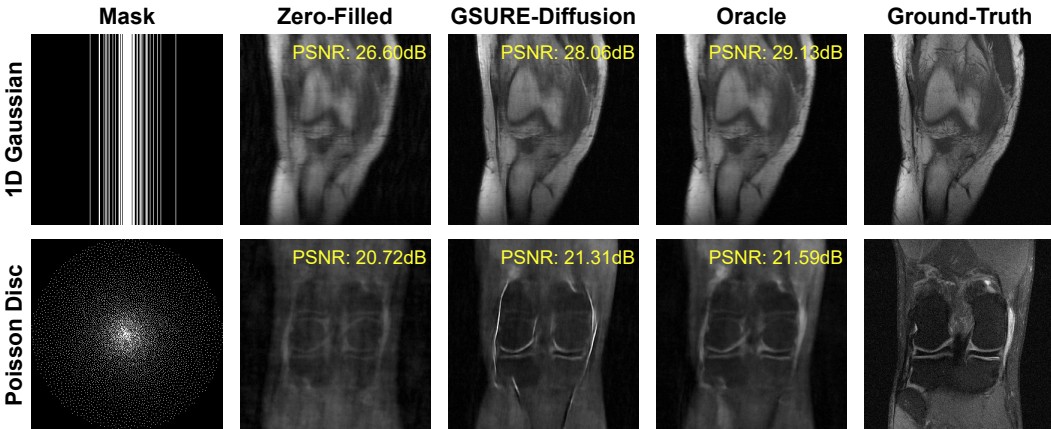

Figure 6: MRI reconstruction results for 1-dimensional Gaussian random sampling ($R = 8$) and variable density Poisson disc sampling ($R = 15$), both with $\sigma_0 = 0.01$. Our method generalizes well for different sampling schemes.

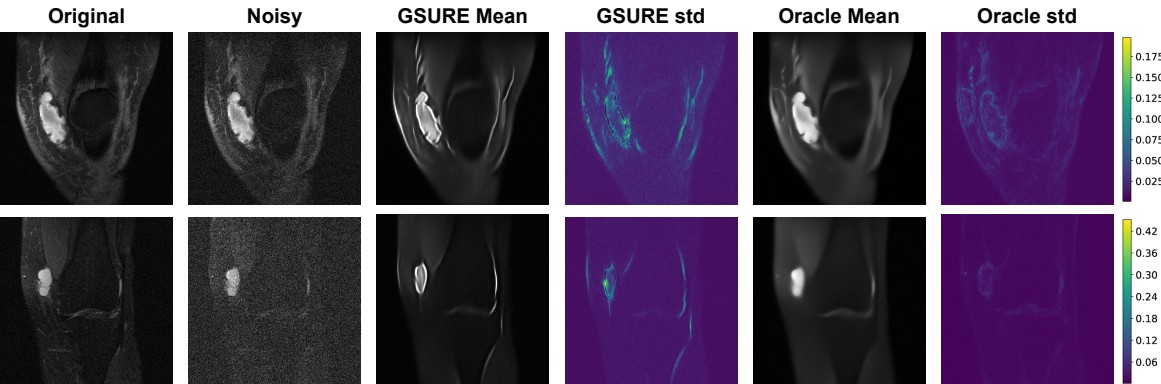

Figure 7: Uncertainty quantification for MR image denoising with GSURE-Diffusion and oracle models. Means and standard deviations are calculated for 8 stochastic diffusion reconstructions.

The latter two limitations can be mitigated in future work by designing a diffusion model architecture that is explicitly aware of the "mask" $P$ for each input, and can handle $\bar{\mathbf{x}}_t$ with a diagonal covariance with different values in the main diagonal. Similar ideas (input masks and diagonal covariance) have been suggested in fully supervised diffusion modeling literature (Bao et al., 2022; Gao et al., 2023), and would be interesting to explore in the corrupted data setting.

# 6  Related Work

There has been a rich vein of works on unsupervised learning from datasets of corrupted data (Lehtinen et al., 2018; Batson & Royer, 2019; Hendriksen et al., 2020; Chen et al., 2022), including several SURE- and GSURE-based approaches (Soltanayev & Chun, 2018; Nguyen et al., 2020; Jo et al., 2021; Metzler et al., 2018; Zhussip et al., 2019; Aggarwal et al., 2022; Abu-Hussein et al., 2022; Liu et al., 2020). While they achieve impressive results, these efforts are mostly focused on learning a specific task. In contrast, our approach learns a foundational generative model, making it suitable for a wide range of applications. While generative modeling methods which learn only using corrupted data have been proposed in the past (Mattei & Frellsen, 2019; Cheng-Xian Li et al., 2019), our method is the first to provide a holistic solution for learning from data corrupted beyond missing pixels, with general linear degradations.

Recent advances in diffusion-based generative modeling (Ho et al., 2020; Dhariwal & Nichol, 2021) have enabled subsequent work to adapt these models for medical imaging (*e.g.*, MRI) (Song et al., 2023; Chung & Ye, 2022; Jalal et al., 2021; Xie & Li, 2022; Chung et al., 2023). These diffusion models can serve a multitude of tasks, but can be expensive to train as they require fully sampled noiseless training data. Notably, DDM² (Xiang et al., 2023) and the concurrent work of SURE-Score (Aali et al., 2023) offer ways to train a diffusion model based on noisy data, which is often the case in practical settings. Similarly, a concurrent work by Daras et al. (2024) offers a method for training diffusion models based on data with missing pixels, showing improved results for downstream inpainting tasks. However, collected data can also be undersampled or corrupted by other transformations. Our proposed framework is more general, as it can handle both Gaussian noise and general linear corruptions.

Diffusion models have permeated various research areas, including online decision making (Hsieh et al., 2023). In that context, obtaining full noiseless data may often be impossible. The authors of (Hsieh et al., 2023) suggest a diffusion loss function that learns a diffusion-based prior from noisy data with missing elements, similar to an image inpainting problem. They note that their proposed loss function could be of independent interest in future work. Our loss function closely resembles theirs, albeit generalizing for linear corruptions beyond inpainting by utilizing the singular value decomposition (SVD).

## 7 Conclusion

We have introduced GSURE-Diffusion, a technique for training generative diffusion models based on data corrupted by linear degradations and additive Gaussian noise. Using the SVD of the degradation operator, GSURE (Eldar, 2008), and an ensemble of degradations (Aggarwal et al., 2022), we introduce a novel training scheme that approximates the underlying data distribution from corrupted measurements. We perform experiments on CelebA (Liu et al., 2015) to demonstrate the validity of our proposed framework. Additionally, we show the applicability of GSURE-Diffusion to real-world problems by training a model on accelerated (undersampled and noisy) MRI scans from fastMRI (Zbontar et al., 2019; Knoll et al., 2020). We then use the resulting diffusion model to address several downstream tasks.

We hope that our proposed framework will enable future work to train generative models on similar problems such as multi-coil accelerated MRI, sparse-view Computed Tomography (CT), and more. Future work may also address additional challenging measurement acquisition scenarios, as dictated by the data modality. We emphasize the fact that our results are obtained on simulated corruptions. More extensive experiments on real data should be conducted before GSURE-Diffusion can be relied upon for diagnostics in clinical settings.

## Acknowledgements

This research was partially supported by the Council For Higher Education - Planning & Budgeting Committee. Data used in the preparation of this article were obtained from the NYU fastMRI Initiative database Zbontar et al. (2019); Knoll et al. (2020). As such, NYU fastMRI investigators provided data but did not participate in analysis or writing of this report. A listing of NYU fastMRI investigators, subject to updates, can be found at fastmri.med.nyu.edu. The primary goal of fastMRI is to test whether machine learning can aid in the reconstruction of medical images.

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
