## A  Proposition Proofs

**Proposition 3.1.** *For* $\mathbf{x} \sim q(\mathbf{x})$, $\bar{\mathbf{x}} = \boldsymbol{V}^\top \mathbf{x}$, $\bar{\mathbf{x}}_t$ *sampled from Equation 8, and the diagonal weight matrix* $\boldsymbol{W} = \mathbb{E}[\boldsymbol{P}]^{-\frac{1}{2}} \succ 0$ *(positive definite), if* $\boldsymbol{P}$ *and* $\left(f_\theta^{(t)}(\bar{\mathbf{x}}_t) - \bar{\mathbf{x}}\right)$ *are statistically independent, then Equation 9 equals Equation 1.*

*Proof.* We focus on the expectation term from Equation 9, which is taken over $\bar{\mathbf{x}}_t \sim q(\bar{\mathbf{x}}_t | \bar{\mathbf{x}}, \boldsymbol{P})$, $\bar{\mathbf{x}} \sim q(\bar{\mathbf{x}})$, $\boldsymbol{P} \sim q_P(\boldsymbol{P})$ with unknown $q(\bar{\mathbf{x}}), q_P(\boldsymbol{P})$. Namely,

$$
\mathbb{E}\left[\left\|\boldsymbol{W}\boldsymbol{P}\left(f_\theta^{(t)}(\bar{\mathbf{x}}_t) - \bar{\mathbf{x}}\right)\right\|_2^2\right]
$$

$$
\overset{1}{=} \mathbb{E}\left[\mathrm{Trace}\left(\boldsymbol{W}\boldsymbol{P}\left(f_\theta^{(t)}(\bar{\mathbf{x}}_t) - \bar{\mathbf{x}}\right)\left(f_\theta^{(t)}(\bar{\mathbf{x}}_t) - \bar{\mathbf{x}}\right)^\top \boldsymbol{P}\boldsymbol{W}\right)\right]
$$

$$
\overset{2}{=} \mathbb{E}\left[\mathrm{Trace}\left(\boldsymbol{P}\boldsymbol{W}^2\boldsymbol{P}\left(f_\theta^{(t)}(\bar{\mathbf{x}}_t) - \bar{\mathbf{x}}\right)\left(f_\theta^{(t)}(\bar{\mathbf{x}}_t) - \bar{\mathbf{x}}\right)^\top\right)\right]
$$

$$
\overset{3}{=} \mathbb{E}\left[\mathrm{Trace}\left(\boldsymbol{W}^2\boldsymbol{P}^2\left(f_\theta^{(t)}(\bar{\mathbf{x}}_t) - \bar{\mathbf{x}}\right)\left(f_\theta^{(t)}(\bar{\mathbf{x}}_t) - \bar{\mathbf{x}}\right)^\top\right)\right]
$$

$$
\overset{4}{=} \mathbb{E}\left[\mathrm{Trace}\left(\boldsymbol{W}^2\boldsymbol{P}\left(f_\theta^{(t)}(\bar{\mathbf{x}}_t) - \bar{\mathbf{x}}\right)\left(f_\theta^{(t)}(\bar{\mathbf{x}}_t) - \bar{\mathbf{x}}\right)^\top\right)\right]
$$

$$
\overset{5}{=} \mathrm{Trace}\left(\mathbb{E}\left[\boldsymbol{W}^2\boldsymbol{P}\left(f_\theta^{(t)}(\bar{\mathbf{x}}_t) - \bar{\mathbf{x}}\right)\left(f_\theta^{(t)}(\bar{\mathbf{x}}_t) - \bar{\mathbf{x}}\right)^\top\right]\right)
$$

$$
\overset{6}{=} \mathrm{Trace}\left(\boldsymbol{W}^2 \mathbb{E}[\boldsymbol{P}]\,\mathbb{E}\left[\left(f_\theta^{(t)}(\bar{\mathbf{x}}_t) - \bar{\mathbf{x}}\right)\left(f_\theta^{(t)}(\bar{\mathbf{x}}_t) - \bar{\mathbf{x}}\right)^\top\right]\right)
$$

$$
\overset{7}{=} \mathrm{Trace}\left(\mathbb{E}\left[\left(f_\theta^{(t)}(\bar{\mathbf{x}}_t) - \bar{\mathbf{x}}\right)\left(f_\theta^{(t)}(\bar{\mathbf{x}}_t) - \bar{\mathbf{x}}\right)^\top\right]\right)
$$

$$
\overset{5}{=} \mathbb{E}\left[\mathrm{Trace}\left(\left(f_\theta^{(t)}(\bar{\mathbf{x}}_t) - \bar{\mathbf{x}}\right)\left(f_\theta^{(t)}(\bar{\mathbf{x}}_t) - \bar{\mathbf{x}}\right)^\top\right)\right]
$$

$$
\overset{8}{=} \mathbb{E}\left[\left\|f_\theta^{(t)}(\bar{\mathbf{x}}_t) - \bar{\mathbf{x}}\right\|_2^2\right]. \tag{11}
$$

Justifications:

1. Using the linear algebra property $\|\mathbf{v}\|_2^2 = \mathrm{Trace}\left(\mathbf{v}\mathbf{v}^\top\right)$ for any vector $\mathbf{v}$, and $\boldsymbol{P} = \boldsymbol{P}^\top, \boldsymbol{W} = \boldsymbol{W}^\top$ because they are diagonal matrices.

2. Using the cyclical shift invariance of the trace operator, $\mathrm{Trace}\left(\boldsymbol{A}\boldsymbol{B}\boldsymbol{C}\right) = \mathrm{Trace}\left(\boldsymbol{C}\boldsymbol{A}\boldsymbol{B}\right)$.

3. Diagonal matrices (such as $\boldsymbol{P}$ and $\boldsymbol{W}$) commute with one another.

4. Since $\boldsymbol{P}$ is a diagonal matrix whose values are either zeroes or ones, it holds $\boldsymbol{P}^2 = \boldsymbol{P}$.

5. For any random matrix $\boldsymbol{A}$, it holds that $\mathbb{E}\left[\mathrm{Trace}\left(\boldsymbol{A}\right)\right] = \mathrm{Trace}\left(\mathbb{E}\left[\boldsymbol{A}\right]\right)$.

6. $\boldsymbol{W}^2$ is a constant and can therefore be taken out of the expectation. Additionally, we use the assumption that the denoiser's error $\left(f_\theta^{(t)}(\bar{\mathbf{x}}_t) - \bar{\mathbf{x}}\right)$ is independent of $\boldsymbol{P}$.

7. $\boldsymbol{W}$ is defined as $\mathbb{E}[\boldsymbol{P}]^{-\frac{1}{2}}$. This results in $\boldsymbol{W}^2\boldsymbol{P} = \boldsymbol{I}$.

8. Using the linear algebra property $\|\mathbf{v}\|_2^2 = \mathrm{Trace}\left(\mathbf{v}\mathbf{v}^\top\right)$ for any vector $\mathbf{v}$.

Equation 11 is identical to the expectation term in Equation 1, except for the distribution of $\bar{\mathbf{x}}_t$ considered in the expectation. Equation 11 considers $\bar{\mathbf{x}}_t \sim q(\bar{\mathbf{x}}_t | \bar{\mathbf{x}}, \boldsymbol{P}) = \mathcal{N}(\sqrt{\bar{\alpha}_t}\boldsymbol{P}\bar{\mathbf{x}}, (1 - \bar{\alpha}_t)\boldsymbol{I})$, whereas Equation 1

considers $\bar{\mathbf{x}}_t \sim q^*(\bar{\mathbf{x}}_t|\bar{\mathbf{x}}) = \mathcal{N}(\sqrt{\bar{\alpha}_t}\bar{\mathbf{x}}, (1-\bar{\alpha}_t)\boldsymbol{I})$. We assume that the neural network $f_\theta^{(t)}(\bar{\mathbf{x}}_t)$ is able to infer $\boldsymbol{P}$ from $\bar{\mathbf{x}}_t$, and can also tailor its output for each $\boldsymbol{P}$ including $\boldsymbol{P} = \boldsymbol{I}$, matching $q^*(\bar{\mathbf{x}}_t|\bar{\mathbf{x}})$. Under these assumptions, both expectations share the same minimizer, thereby completing the proof. A similar proof is presented in ENSURE Aggarwal et al. (2022). □

**Proposition 3.2.** *For* $\mathbf{x} \sim q(\mathbf{x})$, $\bar{\mathbf{x}} = \boldsymbol{V}^\top \mathbf{x}$, $\bar{\mathbf{x}}_t$ *sampled from Equation 8, and* $\lambda_t = 1 - \bar{\alpha}_t$, *it holds that Equation 10 equals Equation 9.*

*Proof.* We utilize a weighted version of the generalized SURE Eldar (2008); Aggarwal et al. (2022) presented in Equation 3,

$$\mathbb{E}\left[\|\boldsymbol{W}\boldsymbol{P}\left(f(\mathbf{y}) - \mathbf{x}\right)\|_2^2\right] = \mathbb{E}\left[\|\boldsymbol{W}\boldsymbol{P}\left(f(\mathbf{y}) - \mathbf{x}_{\mathrm{ML}}\right)\|_2^2\right] + 2\mathbb{E}\left[\nabla_{\boldsymbol{H}^\top \boldsymbol{C}^{-1}\mathbf{y}} \cdot \boldsymbol{W}^2\boldsymbol{P}f(\mathbf{y})\right] + c. \tag{12}$$

This weighted GSURE considers the measurement equation $\mathbf{y} = \boldsymbol{H}\mathbf{x} + \mathbf{z}$ with $\mathbf{z} \sim \mathcal{N}(0, \boldsymbol{C})$, with $\boldsymbol{P} = \boldsymbol{H}^\dagger \boldsymbol{H}$, $\mathbf{x}_{\mathrm{ML}} = \left(\boldsymbol{H}^\top \boldsymbol{C}^{-1}\boldsymbol{H}\right)^\dagger \boldsymbol{H}^\top \boldsymbol{C}^{-1}\mathbf{y}$, and a constant $c$. We consider the measurement equation matching Equation 8, namely $\bar{\mathbf{x}}_t = \sqrt{\bar{\alpha}_t}\boldsymbol{P}\bar{\mathbf{x}} + \bar{\mathbf{z}}_t$ with $\bar{\mathbf{z}}_t \sim \mathcal{N}(0, (1-\bar{\alpha}_t)\boldsymbol{I})$, which is a special case. For these measurements, the left-hand-side in Equation 12 becomes

$$\mathbb{E}\left[\left\|\boldsymbol{W}\left(\sqrt{\bar{\alpha}_t}\boldsymbol{P}\right)^\dagger \left(\sqrt{\bar{\alpha}_t}\boldsymbol{P}\right)\left(f(\bar{\mathbf{x}}_t) - \bar{\mathbf{x}}\right)\right\|_2^2\right] = \mathbb{E}\left[\|\boldsymbol{W}\boldsymbol{P}\left(f(\bar{\mathbf{x}}_t) - \bar{\mathbf{x}}\right)\|_2^2\right],$$

which is identical to the expectation term in Equation 9. This equation holds because $\boldsymbol{P}$ is a diagonal matrix with ones and zeroes, resulting in $\boldsymbol{P}^\dagger = \boldsymbol{P} = \boldsymbol{P}^2$. Meanwhile, by substituting $\boldsymbol{H} = \sqrt{\bar{\alpha}_t}\boldsymbol{P}$, $\boldsymbol{C} = (1-\bar{\alpha}_t)\boldsymbol{I}$, and $\mathbf{y} = \bar{\mathbf{x}}_t$, $\mathbf{x}_{\mathrm{ML}}$ simplifies into

$$\mathbf{x}_{\mathrm{ML}} = \left(\sqrt{\bar{\alpha}_t}\boldsymbol{P}^\top \left((1-\bar{\alpha}_t)\boldsymbol{I}\right)^{-1}\sqrt{\bar{\alpha}_t}\boldsymbol{P}\right)^\dagger \sqrt{\bar{\alpha}_t}\boldsymbol{P}^\top \left((1-\bar{\alpha}_t)\boldsymbol{I}\right)^{-1}\bar{\mathbf{x}}_t$$

$$= \left(\frac{\bar{\alpha}_t}{1-\bar{\alpha}_t}\boldsymbol{P}^\top\boldsymbol{P}\right)^\dagger \frac{\sqrt{\bar{\alpha}_t}}{1-\bar{\alpha}_t}\boldsymbol{P}^\top\bar{\mathbf{x}}_t$$

$$= \frac{1-\bar{\alpha}_t}{\bar{\alpha}_t}\boldsymbol{P}^\dagger \left(\boldsymbol{P}^\top\right)^\dagger \frac{\sqrt{\bar{\alpha}_t}}{1-\bar{\alpha}_t}\boldsymbol{P}^\top\bar{\mathbf{x}}_t$$

$$= \frac{1}{\sqrt{\bar{\alpha}_t}}\boldsymbol{P}\boldsymbol{P}\boldsymbol{P}\bar{\mathbf{x}}_t = \frac{1}{\sqrt{\bar{\alpha}_t}}\boldsymbol{P}\bar{\mathbf{x}}_t.$$

The last two equalities hold because $\boldsymbol{P}^\dagger = \boldsymbol{P}^\top = \boldsymbol{P} = \boldsymbol{P}^2$. Finally, the right-hand-side in Equation 12 becomes

$$\mathbb{E}\left[\left\|\boldsymbol{W}\boldsymbol{P}^\dagger\boldsymbol{P}\left(f(\bar{\mathbf{x}}_t) - \frac{1}{\sqrt{\bar{\alpha}_t}}\boldsymbol{P}\bar{\mathbf{x}}_t\right)\right\|_2^2\right] + 2\mathbb{E}\left[\nabla_{\boldsymbol{P}^\top((1-\bar{\alpha}_t)\boldsymbol{I})^{-1}\bar{\mathbf{x}}_t} \cdot \boldsymbol{W}^2\boldsymbol{P}^\dagger\boldsymbol{P}f(\bar{\mathbf{x}}_t)\right] + c$$

$$\overset{1}{=} \mathbb{E}\left[\left\|\boldsymbol{W}\boldsymbol{P}\left(f(\bar{\mathbf{x}}_t) - \frac{1}{\sqrt{\bar{\alpha}_t}}\bar{\mathbf{x}}_t\right)\right\|_2^2\right] + 2\mathbb{E}\left[\nabla_{\boldsymbol{P}((1-\bar{\alpha}_t)\boldsymbol{I})^{-1}\bar{\mathbf{x}}_t} \cdot \boldsymbol{W}^2\boldsymbol{P}f(\bar{\mathbf{x}}_t)\right] + c$$

$$= \mathbb{E}\left[\left\|\boldsymbol{W}\boldsymbol{P}\left(f(\bar{\mathbf{x}}_t) - \frac{1}{\sqrt{\bar{\alpha}_t}}\bar{\mathbf{x}}_t\right)\right\|_2^2\right] + 2\mathbb{E}\left[\nabla_{(1/(1-\bar{\alpha}_t))\boldsymbol{P}\bar{\mathbf{x}}_t} \cdot \boldsymbol{W}^2\boldsymbol{P}f(\bar{\mathbf{x}}_t)\right] + c$$

$$\overset{2}{=} \mathbb{E}\left[\left\|\boldsymbol{W}\boldsymbol{P}\left(f(\bar{\mathbf{x}}_t) - \frac{1}{\sqrt{\bar{\alpha}_t}}\bar{\mathbf{x}}_t\right)\right\|_2^2\right] + 2\mathbb{E}\left[(1-\bar{\alpha}_t)\nabla_{\boldsymbol{P}\bar{\mathbf{x}}_t} \cdot \boldsymbol{W}^2\boldsymbol{P}f(\bar{\mathbf{x}}_t)\right] + c$$

$$\overset{3}{=} \mathbb{E}\left[\left\|\boldsymbol{W}\boldsymbol{P}\left(f(\bar{\mathbf{x}}_t) - \frac{1}{\sqrt{\bar{\alpha}_t}}\bar{\mathbf{x}}_t\right)\right\|_2^2\right] + 2\mathbb{E}\left[(1-\bar{\alpha}_t)\nabla_{\bar{\mathbf{x}}_t} \cdot \boldsymbol{P}\boldsymbol{W}^2f(\bar{\mathbf{x}}_t)\right] + c$$

$$\overset{4}{=} \mathbb{E}\left[\left\|\boldsymbol{W}\boldsymbol{P}\left(f(\bar{\mathbf{x}}_t) - \frac{1}{\sqrt{\bar{\alpha}_t}}\bar{\mathbf{x}}_t\right)\right\|_2^2 + 2(1-\bar{\alpha}_t)\nabla_{\bar{\mathbf{x}}_t} \cdot \boldsymbol{P}\boldsymbol{W}^2f(\bar{\mathbf{x}}_t) + c\right],$$

which is identical to the expectation term in Equation 10 with $\lambda_t = 1 - \bar{\alpha}_t$. Justifications:

1. $\boldsymbol{P}^\dagger = \boldsymbol{P}^\top = \boldsymbol{P} = \boldsymbol{P}^2$.

2. Using the change of variables formula.

3. Diagonal matrices (such as $\boldsymbol{P}$ and $\boldsymbol{W}$) commute with one another. Additionally, the divergences w.r.t. $\bar{\mathbf{x}}_t$ and w.r.t. $\boldsymbol{P}\bar{\mathbf{x}}_t$ are identical, because $\boldsymbol{P}\boldsymbol{W}^2 f(\bar{\mathbf{x}}_t)$ equals zero in entries where $\boldsymbol{P}$ is zero, and $\boldsymbol{P}\bar{\mathbf{x}}_t$ and $\bar{\mathbf{x}}_t$ are identical in entries where $\boldsymbol{P}$ is non-zero.

4. Using the linearity of the expectation operator.

By rewriting both sides of Equation 12, we obtain that Equation 10 equals Equation 9. $\qquad\square$

## B  Detailed Data Descriptions

### B.1  Dataset Collection

Here, we detail the collection process for the training and testing data in our experiments. Note that the data described here is what we consider pristine uncorrupted data. The corruption process for training GSURE-Diffusion is detailed in subsection B.2.

**CelebA.**  In our experiments on human face images, we use images from the CelebA Liu et al. (2015) dataset. The original CelebA images were center-cropped to $128 \times 128$ pixels, then resized to $32 \times 32$ pixels, and finally turned into grayscale. The images were converted to grayscale by averaging all color channels. Overall, the dataset includes 162770 training set images, and 19867 validation set images (which we use for FID Heusel et al. (2017) evaluations).

**FastMRI.**  We consider all single-coil knee MRI scans from the fastMRI Zbontar et al. (2019); Knoll et al. (2020) dataset, excluding slices with indices below 10 or above 40 as they generally contain less interpretable information. This yields a training set size of 24853. For the validation set we only use the first 1024 valid slices (which we use for all our post-training experiments). We treat each slice as a 2-channel image, separating the complex values into real and imaginary channels. We center-crop the images to a spatial size of $320 \times 320$ following Jalal et al. (2021), and normalize the images by $7e-5$ to obtain better neural network performance. When displaying MR images, we take the absolute value of the complex number in each pixel, and then use min-max normalization to view the resulting values as a grayscale image. In Figure 7, we jointly normalize standard deviations to ensure a fair visual comparison. We attach a color bar to accurately illustrate the standard deviation intensities.

### B.2  Data Corruptions for GSURE-Diffusion

**CelebA.**  In our CelebA Liu et al. (2015) experiments, we consider a degradation operator $\boldsymbol{H}$ that randomly drops each $4 \times 4$-pixel patch with probability $p$. This operator can be mathematically defined as a diagonal matrix $\boldsymbol{H}$ with zeroes in pixels that are dropped and ones in pixels that are kept. The singular value decomposition (SVD) is trivially and efficiently obtained by

$$\boldsymbol{H} = \boldsymbol{IHI}. \tag{13}$$

Note that the singular values in $\boldsymbol{H}$ are not ordered. Since the SVD of $\boldsymbol{H}$ has $\boldsymbol{V}^\top = \boldsymbol{I}$ regardless of the randomness of dropping patches, this family of random operators $\boldsymbol{H}$ matches our assumption that all $\boldsymbol{H}$ share the same left-singular vectors $\boldsymbol{V}^\top$.

Additionally, the projection matrix $\boldsymbol{P} = \boldsymbol{H}^\dagger \boldsymbol{H}$ is simply $\boldsymbol{H}$ (as $\boldsymbol{H}$ is diagonal with zeroes and ones). Because each patch is dropped randomly with probability $p$, it follows that $\mathbb{E}[\boldsymbol{P}] = (1-p)\boldsymbol{I} \succ 0$ is positive definite, matching our assumption.

Finally, we assume $\boldsymbol{H}$ and the additive white Gaussian noise standard deviation $\sigma_0$ to be known for all measurements in the dataset. For simplicity, we assume a uniform $\sigma_0$ for all measurements.

**FastMRI.** For MRI slices from fastMRI Zbontar et al. (2019); Knoll et al. (2020), the degradation operator we use is the horizontal frequency subsampling operator used in Jalal et al. (2021). For an acceleration factor $R$, the degradation operator $\boldsymbol{H}$ keeps the central $120/R$ frequencies, and then uniformly samples an additional $200/R$ frequencies. This results in a sampling of $320/R$ frequencies out of the original 320. More formally,

$$\boldsymbol{H} = \boldsymbol{I\Sigma F}, \tag{14}$$

where $\boldsymbol{F}$ is the discrete Fourier transform matrix, and $\boldsymbol{\Sigma}$ is a square diagonal matrix containing ones for frequency indices that are kept by $\boldsymbol{H}$, and zeroes elsewhere. Incidentally, Equation 14 is a valid SVD of $\boldsymbol{H}$, and can be efficiently simulated using the fast Fourier transform algorithm.

This operator matches our assumptions: (i) We assume each $\boldsymbol{H}$ and the additive white Gaussian noise standard deviation $\sigma_0$ to be known; (ii) All matrices $\boldsymbol{H}$ share the same left-singular vectors defined by $\boldsymbol{F}$ (and not depending on the randomness); and (iii) The central $120/R$ horizontal frequencies are always sampled, and each of the remaining frequencies are equally likely to be sampled, with probability $200/(320R - 120)$. Thus $\mathbb{E}[\boldsymbol{P}] = \mathbb{E}[\boldsymbol{\Sigma}^\dagger \boldsymbol{\Sigma}]$ is a diagonal matrix with nonzero diagonal values, making it positive definite.

## C   Implementation Details

Our experiments were conducted using DDPM Ho et al. (2020) U-Net architecture with base channel width 128. All networks were trained using the Adam optimizer, dropout with probability 0.1, EMA with decay factor of 0.9999. The diffusion process considered in training for all experiments has 1000 timesteps, with a linear $\beta$ schedule ranging from $\beta_1 = \sigma_0^2$ ($\sigma_0^2$ is the variance of the AWGN in the data) to $\beta_{1000} = 0.2$. All experiments were conducted on 8 NVIDIA A40 GPUs.

In the human faces experiment we ignore the weighting matrix $\boldsymbol{W}$ during training because the probability for each pixel to be masked is uniform. For the knee MRI experiment the weighting matrix $\boldsymbol{W}$ was set to 1 for the central lines that were not masked by $\boldsymbol{H}$, and $\sqrt{5.8}$ for all other lines matching their inverse square root masking probability (for $R = 4$).

All models, including the oracle ones, were trained with the hyperparameters listed in Table 2. The "mean type" hyperparameter refers to whether the neural network predicts the image $\mathbf{x}$ or the added noise $\boldsymbol{\epsilon} = \left(\mathbf{x}_t - \sqrt{\bar{\alpha}_t}\mathbf{x}\right) / \left(\sqrt{1 - \bar{\alpha}_t}\right)$.

Table 2: Architecture and training hyperparameters for CelebA Liu et al. (2015) and fastMRI Zbontar et al. (2019); Knoll et al. (2020) experiments.

|  | CelebA | FastMRI |
|---|---|---|
| **Iterations** | 180,000 | 31,000 |
| **Batch Size** | 128 | 32 |
| **Learning Rate** | $5e-5$ | $1e-5$ |
| **Mean Type** | `predict_x` | `predict_epsilon` |
| **Channel Multipliers** | $[1, 2, 2, 2, 4]$ | $[1, 1, 2, 2, 4, 4]$ |
| **Attention Resolutions** | $[16]$ | $[20]$ |
| $\gamma_t$ | 1 | $\frac{\bar{\alpha}_t}{1-\bar{\alpha}_t}$ |
| $\lambda_t$ | 0.0001 | $0.0001 \cdot \frac{1-\bar{\alpha}_t}{\bar{\alpha}_t}$ |

For the MRI model, we apply an inverse Fourier transform and a Fourier transform to the network's input and output respectively, to utilize the convolutional architecture's advantage on image data (rather than frequencies). Due to the orthogonality and linearity of the Fourier transform and its inverse, the additive white Gaussian noise remains so, and maintains the same variance.

Algorithm 1 shows the GSURE-Diffusion training algorithm used in this paper. We provide our anonymized code and configuration files in the supplementary material. We intend to publish the code along with our trained model checkpoints upon acceptance.

---

**Algorithm 1:** (**V3qg**) GSURE-Diffusion training algorithm

---

**Input:** a dataset $\mathcal{D}$ of corrupted images $\mathbf{y}$ with known degradation matrices $\mathbf{H}$ and noise amplitudes $\sigma_0$, learning rate $\eta$, hyperparameters $\lambda_t, \bar{\alpha}_t, T$, and DNN $f_\theta^{(t)}(\bar{\mathbf{x}}_t)$ with initial parameters $\theta$.

**1** Initialize $\bar{\mathcal{D}} \leftarrow \{\}$                                                    `// precompute measurements before training`

**2** **for** $\mathbf{y}, \mathbf{H}, \sigma_0$ *in* $\mathcal{D}$

**3**      $\mathbf{U}, \boldsymbol{\Sigma}, \mathbf{V}^\top \leftarrow \texttt{SVD}(\mathbf{H})$

**4**      $\mathbf{P} \leftarrow \boldsymbol{\Sigma}^\dagger \boldsymbol{\Sigma}$

**5**      $\bar{\mathbf{y}} \leftarrow \boldsymbol{\Sigma}^\dagger \mathbf{U}^\top \mathbf{y}$

**6**      store $\bar{\mathbf{y}}, \boldsymbol{\Sigma}, \mathbf{P}, \sigma_0$ in $\bar{\mathcal{D}}$

**7** $\mathbf{W} \leftarrow \mathbb{E}_{\mathbf{P} \sim \bar{\mathcal{D}}}[\mathbf{P}]$

**8** **for** $N$ *epochs*                                                           `// training loop`

**9**      **for** $\bar{\mathbf{y}}, \boldsymbol{\Sigma}, \mathbf{P}, \sigma_0$ *in* $\bar{\mathcal{D}}$

**10**          $t \sim \mathcal{U}[1, T], \boldsymbol{\epsilon}_t \sim \mathcal{N}(0, \mathbf{I})$

**11**          $\bar{\mathbf{x}}_t \leftarrow \sqrt{\bar{\alpha}_t}\bar{\mathbf{y}} + \left((1 - \bar{\alpha}_t)\mathbf{I} - \bar{\alpha}_t\sigma_0^2\boldsymbol{\Sigma}^\dagger\boldsymbol{\Sigma}^{\dagger\top}\right)^{\frac{1}{2}}\boldsymbol{\epsilon}_t$

**12**          $\mathcal{L}(\theta) \leftarrow \left\|\mathbf{WP}\left(f_\theta^{(t)}(\bar{\mathbf{x}}_t) - \bar{\mathbf{y}}\right)\right\|_2^2 + 2\lambda_t\left(\nabla_{\bar{\mathbf{x}}_t} \cdot \mathbf{PW}^2 f_\theta^{(t)}(\bar{\mathbf{x}}_t)\right)$

**13**          $\theta \leftarrow \theta - \eta\nabla_\theta\mathcal{L}(\theta)$

**Output:** $\theta$                                                            `// trained network parameters`

---

## D   Pragmatic Loss Function Considerations

### D.1   Divergence Term Estimation

The GSURE-Diffusion training loss in Equation 10 contains a divergence term, which is highly expensive to accurately obtain, in both memory consumption and computation time. Similar to other SURE-based methods Metzler et al. (2018); Soltanayev & Chun (2018); Aggarwal et al. (2022), we use an unbiased Monte Carlo approximation Ramani et al. (2008) of the divergence. Considering the divergence as the trace of the Jacobian matrix $\boldsymbol{J}$ of the term being differentiated $(\boldsymbol{PW}^2 f_\theta^{(t)}(\bar{\mathbf{x}}_t))$, Monte Carlo SURE Ramani et al. (2008) uses Hutchinson's trace estimator Hutchinson (1989). We compute the estimate by sampling a random Gaussian vector $\mathbf{v} \sim \mathcal{N}(0, \boldsymbol{I})$, and calculating $\mathbf{v}^\top \boldsymbol{J} \mathbf{v}$ using automatic differentiation tools. Notably, this differs from previous methods Metzler et al. (2018); Soltanayev & Chun (2018); Aggarwal et al. (2022) that used numerical estimates for differentiation, which may suffer from numerical inaccuracies.

### D.2   MSE Term Variance

The GSURE-Diffusion loss function in Equation 10 contains the following squared error term

$$\left\|\boldsymbol{WP}\left(f_\theta^{(t)}(\bar{\mathbf{x}}_t) - \frac{1}{\sqrt{\bar{\alpha}_t}}\bar{\mathbf{x}}_t\right)\right\|_2^2.$$

Because of the possibly strong noise-to-signal ratio present in $\bar{\mathbf{x}}_t$, this term may suffer from high variance, effectively impeding the training process. To alleviate this, we propose replacing $\frac{1}{\sqrt{\bar{\alpha}_t}}\bar{\mathbf{x}}_t$ with the less noisy $\bar{\mathbf{y}}$, resulting in the loss function

$$\sum_{t=1}^T \gamma_t\mathbb{E}\left[\left\|\boldsymbol{WP}\left(f_\theta^{(t)}(\bar{\mathbf{x}}_t) - \bar{\mathbf{y}}\right)\right\|_2^2 + 2\lambda_t\left(\nabla_{\bar{\mathbf{x}}_t} \cdot \boldsymbol{PW}^2 f_\theta^{(t)}(\bar{\mathbf{x}}_t)\right) + c\right]. \tag{15}$$

Note that the difference between the expectations in Equation 10 and Equation 15 is negligible, while Equation 15 has significantly less variance (as $\bar{\mathbf{y}}$ is less noisy than $\frac{1}{\sqrt{\bar{\alpha}_t}}\bar{\mathbf{x}}_t$). From Equation 7 we get

$$\frac{1}{\sqrt{\bar{\alpha}_t}}\bar{\mathbf{x}}_t = \bar{\mathbf{y}} + \frac{1}{\sqrt{\bar{\alpha}_t}}\left((1 - \bar{\alpha}_t)\boldsymbol{I} - \bar{\alpha}_t\sigma_0^2\boldsymbol{\Sigma}^\dagger\boldsymbol{\Sigma}^{\dagger\top}\right)^{\frac{1}{2}}\boldsymbol{\epsilon}_t.$$

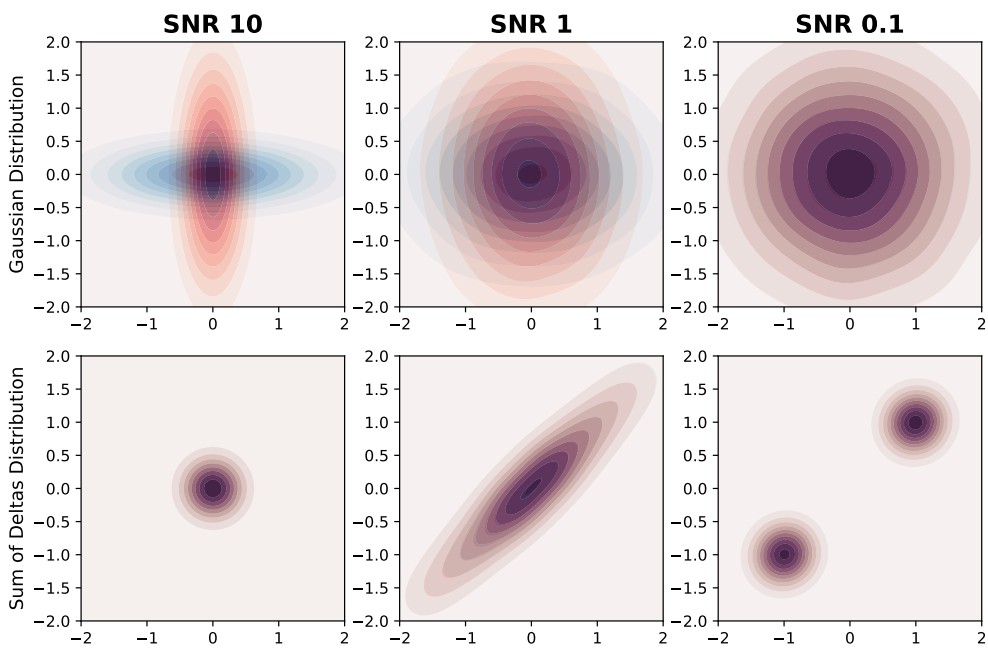

Figure 8: Analytic example of independence assumptions. The two-dimensional distributions depicted in the graph consist of an isotropic Gaussian distribution (top) and a sum of two delta functions with equal probabilities centered around (1, 1) and (-1, -1) respectively (bottom). The graph illustrates the KDE of the error vectors from the denoiser, with two distinct colors representing two different types of data-point degradations. The blue color represents the denoiser error when the first element is masked, while the red color indicates the denoiser error when the last element is masked. At a low SNR, the KDE of both distributions appears identical when masking is applied to either element. As SNR increases, this independence is weakened in the isotropic Gaussian setting, but remains for the sum of deltas distribution.

We denote $\bar{\boldsymbol{\epsilon}} = \frac{1}{\sqrt{\bar{\alpha}_t}} \left( (1 - \bar{\alpha}_t) \boldsymbol{I} - \bar{\alpha}_t \sigma_0^2 \boldsymbol{\Sigma}^\dagger \boldsymbol{\Sigma}^{\dagger\top} \right)^{\frac{1}{2}} \boldsymbol{\epsilon}_t$, and show that

$$\left\| \boldsymbol{W}\boldsymbol{P} \left( f_\theta^{(t)}(\bar{\mathbf{x}}_t) - \frac{1}{\sqrt{\bar{\alpha}_t}} \bar{\mathbf{x}}_t \right) \right\|_2^2$$
$$= \left\| \boldsymbol{W}\boldsymbol{P} \left( f_\theta^{(t)}(\bar{\mathbf{x}}_t) - \bar{\mathbf{y}} - \bar{\boldsymbol{\epsilon}} \right) \right\|_2^2$$
$$= \left\| \boldsymbol{W}\boldsymbol{P} \left( f_\theta^{(t)}(\bar{\mathbf{x}}_t) - \bar{\mathbf{y}} \right) \right\|_2^2 + \| \boldsymbol{W}\boldsymbol{P}\bar{\boldsymbol{\epsilon}} \|_2^2 - 2\bar{\boldsymbol{\epsilon}}^\top \boldsymbol{P}\boldsymbol{W}\boldsymbol{W}\boldsymbol{P} \left( f_\theta^{(t)}(\bar{\mathbf{x}}_t) - \bar{\mathbf{y}} \right)$$
$$= \left\| \boldsymbol{W}\boldsymbol{P} \left( f_\theta^{(t)}(\bar{\mathbf{x}}_t) - \bar{\mathbf{y}} \right) \right\|_2^2 - 2\bar{\boldsymbol{\epsilon}}^\top \boldsymbol{P}\boldsymbol{W}^2\boldsymbol{P} f_\theta^{(t)}(\bar{\mathbf{x}}_t) + 2\bar{\boldsymbol{\epsilon}}^\top \boldsymbol{P}\boldsymbol{W}^2\boldsymbol{P}\bar{\mathbf{y}} + \| \boldsymbol{W}\boldsymbol{P}\bar{\boldsymbol{\epsilon}} \|_2^2 .$$

The final two terms are constants w.r.t. $\theta$. Effectively, this means that the difference between the squared error terms in Equation 10 and Equation 15 is $2\bar{\boldsymbol{\epsilon}}^\top \boldsymbol{P}\boldsymbol{W}^2\boldsymbol{P} f_\theta^{(t)}(\bar{\mathbf{x}}_t)$. Under the manifold hypothesis, if $f_\theta^{(t)}(\bar{\mathbf{x}}_t)$ outputs valid images residing on the manifold, and because $\bar{\boldsymbol{\epsilon}}$ is a random Gaussian vector, $f_\theta^{(t)}(\bar{\mathbf{x}}_t)$ and $\bar{\boldsymbol{\epsilon}}$ are perpendicular. Therefore, the expected difference between the squared error terms, $\mathbb{E}\left[ 2\bar{\boldsymbol{\epsilon}}^\top \boldsymbol{P}\boldsymbol{W}^2\boldsymbol{P} f_\theta^{(t)}(\bar{\mathbf{x}}_t) \right]$, is zero. This motivates us to replace Equation 10 with Equation 15, resulting in significantly lower variance in the loss at little to no cost in terms of bias.

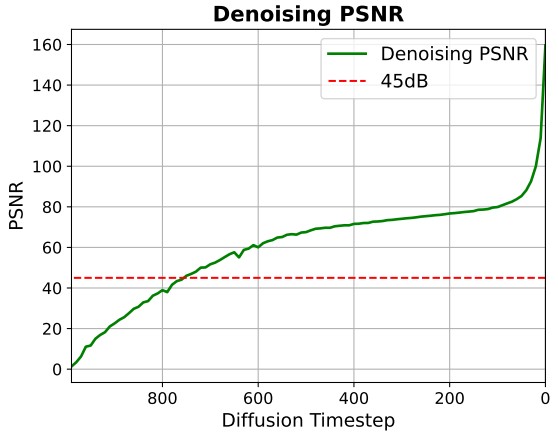
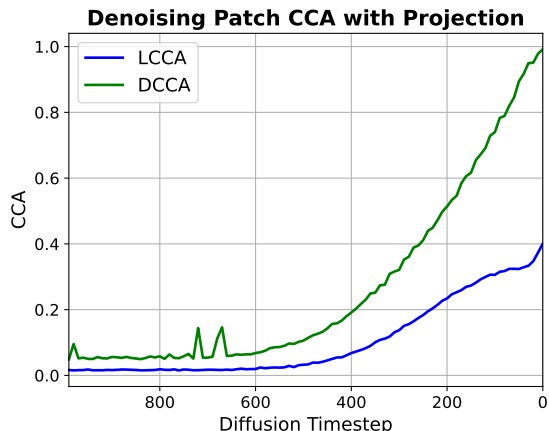

(a) Denoising PSNR between the output of an oracle model and a model trained with GSURE-Diffusion on noisy uncorrupted images.

(b) Assessment of correlation between the denoising error and projection, using linear and deep canonical correlation analysis.

Figure 9: Empirical assessment of assumptions.

### D.3 Assessment of Generalization Assumption

According to Proposition 3.1, the GSURE-Diffusion loss function is equivalent to an MSE objective for samples $\bar{\mathbf{x}}_t \sim \mathbb{E}_{\boldsymbol{P}}[q(\bar{\mathbf{x}}_t|\bar{\mathbf{x}},\boldsymbol{P})|\boldsymbol{P}]$. During inference, we assume the network is capable of denoising samples from $\bar{\mathbf{x}}_t \sim q^*(\bar{\mathbf{x}}_t|\bar{\mathbf{x}})$. To asses the generalization capabilities of the network on such samples, we compare the output of a network trained with GSURE-Diffusion and with an oracle network trained on uncorrupted data. In Figure 9a we present the PSNR between the network outputs for denoising samples from $\bar{\mathbf{x}}_t \sim q^*(\bar{\mathbf{x}}_t|\bar{\mathbf{x}}) = q(\bar{\mathbf{x}}_t|\bar{\mathbf{x}},\boldsymbol{P}=\boldsymbol{I})$. The graph shows that for most time-steps, the PSNR is higher than 45dB, leading us to conclude that the model trained with GSURE-Diffusion is an adequate denoiser for samples from $\bar{\mathbf{x}}_t \sim q^*(\bar{\mathbf{x}}_t|\bar{\mathbf{x}})$, and the assumption is valid.

### D.4 Assessment of Independence Assumption

Our proof for Proposition 3.1 makes use of the assumption made in ENSURE (Aggarwal et al., 2022), that the network's denoising error is independent of the projection matrix $\boldsymbol{P}$. To provide some insight into the limitations of this assumption, we begin with a simple example of two-dimensional distributions; one an isotropic multivariate Gaussian and the other a sum of two delta functions, sampled with equal probabilities. From each distribution we sample 10,000 samples $\bar{\mathbf{x}}_t$ using Equation 7, where our degradation $\boldsymbol{P}$ randomly masks one of the two coordinates with equal probability. We then compute the error $\left(f_\theta^{(t)}(\bar{\mathbf{x}}_t) - \bar{\mathbf{x}}\right)$ for the two possible projections and examine whether the error is independent of the projection $\boldsymbol{P}$. The probabilities of the error vector are depicted in Figure 8, using kernel density estimation (KDE). From our analysis, we draw the following observations: In scenarios with high noise levels, the error probabilities are identical for either projection $\boldsymbol{P}$, regardless of the distribution. This indicates that much of the information from the original data-point is lost due to the noise, rendering the projection largely ineffectual. For low noise values, the two distributions diverge. The error vectors created from points in the Gaussian distributions are highly correlated with $\boldsymbol{P}$, while the ones created from the two deltas remain indistinguishable. Error vectors generated from points in the Gaussian distribution are highly correlated with $\boldsymbol{P}$, whereas those from the two delta functions remain indistinguishable. Hence, we infer that the assumption of independence between the network's denoising error and the projection matrix $\boldsymbol{P}$ is dependent upon the underlying dataset and degradation family, necessitating empirical validation before applying our method.

We assess the validity of this assumption for our trained networks using Canonical Correlation Analysis (CCA) measured on corresponding $8 \times 8$ patches from the denoising error and the projection matrix $\boldsymbol{P}$. We

perform this comparison using the model trained in subsection 4.1, where we can assume locality of the correspondence as the projection matrix $P$ is an inpainting binary mask. The model and noise schedule used for the CelebA (Liu et al., 2015) assessment fit the model trained with $p = 0.2$, $\sigma_0 = 0.01$. We present the results in Figure 9b, showing both linear and deep CCA (LCCA and DCCA accordingly). As shown in the graphs, the assumption made in ENSURE (Aggarwal et al., 2022) nearly holds for our model for timesteps above 500. The assumption grows less accurate the lower the timestep used in training. This inaccuracy suggests that there may be room for improvement in the assumptions made in ENSURE (Aggarwal et al., 2022), which we believe may be interesting for future work.

## E   Additional Results

We repeat the human face experiment in subsection 4.1 with color images from CelebA64 (Liu et al., 2015). Both models are trained with the parameters listed in Appendix C for the human face experiment, however the models were trained for a longer 250,000 iterations corresponding to the higher dimension of the data. The finding, shown in Table 3, from the 10,000 image FID (Heusel et al., 2017) with the validation set reflect those in subsection 4.1.

Table 3: FID (Heusel et al., 2017) results for diffusion models trained degraded data for color $32 \times 32$-pixel CelebA (Liu et al., 2015) images, with different DDIM (Song et al., 2020a) steps at generation time. Models were trained with (top) or without (bottom) GSURE-Diffusion, on degraded data.

| Training Scheme | Data Degradation | 10 Steps | 20 Steps | 50 Steps | 100 Steps |
|---|---|---|---|---|---|
| Regular | No degradation (oracle) | 17.40 | 11.18 | 08.32 | 07.21 |
| GSURE-Diffusion | $p = 0.2$, $\sigma_0 = 0.01$ | 17.39 | 11.89 | 09.31 | 08.97 |