# OpenReview forum: "GSURE-Based Diffusion Model Training with Corrupted Data"
_TMLR — Accepted by TMLR_

### Review · Reviewer_HXZD · 2023-12-02

**Summary Of Contributions:**

This paper introduces GSURE-Diffusion, a new training method for generative diffusion models based only on data corrupted by linear degradations and Gaussian noise. It employs a loss function derived from the Generalized Stein’s Unbiased Risk Estimator (GSURE). The paper offers both theoretical guarantees for the loss function and empirical validations of the method.

**Audience:**

Yes

**Claims And Evidence:**

Yes

**Requested Changes:**

In sec 3.2, the authors assume that “The estimate assumes that the model has the ability to generalize for samples from..despite having trained only on signals with an undersampled P” I wonder how this can be verified, and are there any conditions (on date distribution and P) for this?

In prop 3.1, is W a deterministic weight matrix? If it is, then eq(9) can be simplified or rewritten to emphasize this. The current formulation of eq(9) looks like that W is also random.

The assumption needed in prop 3.1., i.e., “network’s denoising error is independent of the projection matrix P,” was assessed using Canonical Correlation Analysis (CCA) in the appendix. I am a bit confused about the assumption related to data generation, specifically how the projection matrix P, which represents missing values, is selected in practice. For instance, if the missing scheme implies that P does not depend on x (I think this is the case for sec 4.1), does this naturally imply independence in the assumption? On the other hand, if P depends on data (meaning missing not completely at random), can we still expect such independence?

The setting in Fig 1 is similar to the case with missing values (although here the authors also impose Gaussian noise). I wonder how would the authors compare with generative models when missing values are present. To name a few:

Mattei, Pierre-Alexandre, and Jes Frellsen. "MIWAE: Deep generative modelling and imputation of incomplete data sets." International conference on machine learning. PMLR, 2019.

Li, Steven Cheng-Xian, Bo Jiang, and Benjamin Marlin. "MisGAN: Learning from Incomplete Data with Generative Adversarial Networks." International Conference on Learning Representations. 2018.

Minor:
- Figure ?? in sec 3.2 is missing.
- The third paragraph of the introduction is a bit hard to follow, especially for readers not familiar with SVD-transformed measurements, Generalized Stein’s Unbiased Risk Estimator, supervised denoising diffusion loss, etc. I suggest to rewrite it a bit to give better explanations and summaries of the proposed approach.

**Strengths And Weaknesses:**

Strength:
1. Training diffusion models with corrupted data is an important research problem, and of great importance for real applications.
2. This paper incorporates the GSURE into the diffusion training, which improves the performance of the Regular diffusion on corrupted data.

Weakness:
1. Some key assumptions are not clearly justified or explained; see details in the comments below.
2. The matrix P closely resembles, if not exactly interpreted as, the masks for missing values. However, the paper lacks discussion or comparison with works that focus on training generative models with missing values, making it challenging to judge the specific novelty of this work.

---

### Review · Reviewer_vrwZ · 2023-12-11

**Summary Of Contributions:**

This paper proposes GSURE-Diffusion, a method for learning a generative model from corrupted data. The form of corruption is general, e.g. it could be linear corruption or Gaussian noises.

Formally, the method is based on GSURE (generalize SURE). Let the corrupted measurements be $y := Hx + z$ for some degradation procedure $H$, the paper defines a diagonal subsampling matrix $P := \Sigma^\dagger \Sigma$ where $\Sigma$ is given by the SVD of $H$. The proposed objective is defined based on $P$.
- The paper theoretically shows that the proposed objective is equivalent to the fully supervised (i.e. without subsampling) denoising objective (Prop 3.1 & 3.2).
- The objective is future modified (i.e. eq (10) to eq (15)) to reduce the variance in estimation.

Empirically, the paper provides results on CelebA and FastMRI.
- For both dataset, the paper provides the specific choices of $H$.
- On FastMRI, the paper demonstrates that the method can:
  - generalize to higher acceleration factors; for example, it's trained on R=4 and can generalize to R = 6,8,10,12.
  - generalize to other types of masks; for example, it's trained on (todo) and can generalize to 1d Gaussian masks or Poisson masks.
  - be used for uncertainty quantification, reported spatially.

**Audience:**

Yes

**Broader Impact Concerns:**

There are no ethical concerns.
For broader impact, this work may be used for medical settings, where uncertainty quantification is a desideratum.
The paper has a short discussion on potentially providing uncertainty metrics, but more discussions / considerations are needed (as mentioned in my questions above).

**Claims And Evidence:**

Yes

**Requested Changes:**

I'd like to have some clarifications please:
- Appendix D.3: when checking for generalization of an undersampled $P$ to the identity $I$, why is 45dB chosen as a threshold for PSNR?
- Table 1: how many runs are there for each setup?
- Fig 5: why do all masks have a masked out strip at the center?
- Fig 7: it's unclear how the uncertainty estimates could be helpful. Currently it seems that the uncertainty is high in non-black regions, which however are regions containing information (since black is the background).

Changes:
- Please add more datasets, preferably with at least one dataset with colored images.
- Please add comparisons to other methods (e.g. in prior work), rather than just the oracle.
- Please consider a more thorough literature review, e.g. more discussions on prior methods on inverse problems.

**Strengths And Weaknesses:**

Strengths:
- The validity of the proposed objective is theoretically justified (Prop 3.1 & 3.2).
- The empirical results are promising, comparable with the oracle (i.e. full measurement) results on various metrics.
- The paper clearly discusses limitations:
    - the data assumptions may not be satisfied in real-world datasets.
    - the performance may suffer if the noise is high variance (in some coordinates).
    - the performance may suffer if different parts of the distribution contribute unevenly to the subsampling matrix $P$.

Weaknesses / limitations:
- For empirical results, my main concern is that the method has limited comparison and evaluation:
  - The model is evaluated on CelebA and FastMRI, both are small datasets of grayscale images. Results on more complex data will be needed to better evaluate the of the proposed method.
  - The proposed method is compared with the oracle only, and there are no comparison with methods in prior work. I apologize that I'm not an expert in this field so am not able to provide specific pointers on which papers to compare to. However, I think the related work section should be much more thorough, given the large body of work in diffusion models / image denoising.

---

### Review · Reviewer_6gUc · 2024-02-06

**Summary Of Contributions:**

The manuscript suggests training a diffusion model with only partial and noisy observations, $y_i = H_i x_i + \textrm{(noise)}$. Here, the observation operator $H$ varies but is known, and can be represented as $H_i = U_i^\top \Sigma_i V$, where $V$ is a constant orthonormal matrix. This setup allows reformulating the problem using known orthogonal projection operators $P_i$, as $\overline{y}_i = P_i , \overline{x}_i + \textrm{(noise)}$.

For the diffusion model to be effectively trained, a diverse set of projections $P_i$ is required. The paper introduces a strategy to adapt the conventional training method for diffusion models to this context. A key innovation is replacing the standard mean squared error (MSE) metric, $||\textrm{(denoiser estimate)} - \text{(truth)}||^2$, with an unbiased estimate derived from Stein's Lemma.

The manuscript presents two set of experiments that demonstrate the effectiveness of the method.

**Audience:**

Yes

**Broader Impact Concerns:**

no concern.

**Claims And Evidence:**

Yes

**Requested Changes:**

1. I have found the text very interesting. I am still not understanding well the point I am raising in the "weakness" section above regarding the condition that $\mathbb{E}[P]$ is positive definite is enough for learning the distribution of clean samples. In any case, I think that this should be clarified more clearly in the manuscript.
2. While not entirely crucial, I still think that running a few (easy and low-dimensional) experiments in a tractable setting as described in the previous section to be worth it to understand the method better.

**Strengths And Weaknesses:**

As a disclaimer, I would like to stress that I am not on top of the (very large) recent literature on diffusion models.

## Strength:
1. The manuscript is well-written and mostly straightforward. The addressed problem—training a diffusion model with degraded samples—is highly relevant, with significant practical implications, notably in medical imaging.  As a non-expert, I have found the literature review adequate for quickly placing the manuscript within the current research landscape.
2. From my assessment, the proposed method seems to be the first to train diffusion models using degraded samples in the form $H_i x_i + \textrm{(noise)}$. The most similar existing work is [1], which only deals with additive Gaussian noise (i.e., $H_i = Id$).
3. The numerical experiments are robust, and the empirical results are compelling.

## Weaknesses & Questions:
1. In equation (7), why is the term $(1-\alpha_t) I - \alpha_t  \sigma_0^2 \Sigma^\dagger \Sigma$ semi-definite positive?
2. One of the conditions for the approach to work is for the set of different $H$ to "cover" the signal space, which is formulated as the condition that $\mathbb{E}[P]$ is positive definite. However, I'm unsure why this condition is considered sufficient. Take the scenario where samples are in $\mathbb{R}^{2D}$, represented as $[x1,x2]$ with $x_1, x_2 \in \mathbb{R}^{D}$, and there are only two possible observation operators: $H_1[x_1, x_2] = x_1$ and $H_2[x_1, x_2] = x_2$. Essentially, we observe either the first $D$ components or the last $D$ components. Here, the matrices associated are $P_1=H_1$ and $P_2=H_2$, and $V=I$, meeting the $\mathbb{E}[P]$ positive definiteness criterion. Yet, it's unclear how one could estimate the joint distribution of $[x_1, x_2]$—particularly, how to infer the correlation structure between the first and last $D$ components when they're never (noisily) observed together.
3. divergence term is computed by the Hutchinson estimator: some comments on this estimator that is often extremely noisy wold be appreciated.
4. this would have been, I think, extremely interesting to run a few simulations on very low dimensional, possibly correlated Gaussian, examples. For example, why not take the clean samples generated from a low-dimensional Gaussian distribution with known covariance structure, and use some simple observation operator such as "observation of a few coordinates"-- this setting is entirely tractable. This would allow one to understand a bit more clearly the different trade-offs that are at play.

### Minor issues:
1. [Figure ??] --> broken ref page 5
2. page 5: "$\frac{1}{ \sqrt{\overline{\alpha}_t} } \overline{x}_t$ is placed by $\overline{y}$": could you please give mode comments and details.






[1]: Aali, Asad, et al. "Solving Inverse Problems with Score-Based Generative Priors learned from Noisy Data." arXiv preprint arXiv:2305.01166 (2023).

---

### Decision · Action_Editor_Rzse · 2024-04-07

**Recommendation:** Accept with minor revision

**Comment:**

Please include the latest recommendation of the reviewers:  it may be helpful to include a discussion on [1], which might have been a concurrent work.
[1] Giannis Daras, Kulin Shah, Yuval Dagan, Aravind Gollakota, Alexandros G. Dimakis, Adam Klivans. Ambient Diffusion: Learning Clean Distributions from Corrupted Data

The paper is fine otherwise.

**Audience:**

It is interesting for paper working on inverse problems for imaging (medical imaging in particular).

**Claims And Evidence:**

The manuscript introduces a novel method for training diffusion models by replacing the standard MSE loss with an unbiased estimate derived from Stein's Lemma. This work may be quite impactful in applications such as medical imaging
This proposed solution seems is overall useful --- it is relatively straightforward to implement.